# Disentangling the influence of ecological and historical factors on seed germination and seedling types in a Neotropical dry forest

Jorge Cortés-Flores[1,2¤], Guadalupe Cornejo-Tenorio[1☯], María Esther Sánchez-Coronado[3☯], Alma Orozco-Segovia[3☯], Guillermo Ibarra-Manríquez[1☯]*

1 Instituto de Investigaciones en Ecosistemas y Sustentabilidad, Universidad Nacional Autónoma de México, Morelia, Michoacán, México, 2 Laboratorio Nacional de Análisis y Síntesis Ecológica, Escuela Nacional de Estudios Superiores Unidad Morelia, Universidad Nacional Autónoma de México, Morelia, Michoacán, México, 3 Departamento de Ecología Funcional, Instituto de Ecología, Universidad Nacional Autónoma de México, Ciudad de México, México

☯ These authors contributed equally to this work.
¤ Current address: Jardín Botánico, Instituto de Biología, sede Tlaxcala, Ex Fábrica San Manuel S/N, Santa Cruz Tlaxcala, Tlaxcala, México
* gibarra@cieco.unam.mx

**Data Availability Statement:** All relevant data are within the manuscript and its Supporting Information files.

## Abstract

In tropical dry forests, although seed germination and seedling establishment are in general limited by the seasonal availability of water, high interspecific variability, nonetheless, exists in terms of seedling traits and germination dynamics. Differences among species in seed germination and seedling traits may be related to other plant life-history traits, such that assessing these relationships may increase our understanding of factors influencing plant establishment, which would affect the regeneration pathways of tropical dry forest communities. In this study, taking into consideration the effect species' phylogeny, we evaluated the relationships of seed germination metrics (percentage, lag time, and rate of germination) and seedling types (i.e. cotyledons functional morphology), with plant life-history traits (growth form, seed mass, dispersal syndrome and dispersal phenology) for 110 species in a Neotropical dry forest in Mexico. A total of 92% of the species studied disperse their seeds during the dry season, mainly at the beginning of this season (66%), a strategy mostly associated with autochorous herbs. Seed germination was more frequent in species that dispersed seeds at the end of the dry season. Germination percentage was not related to any of the traits studied. However, germination lag time and rate were negatively related to seed mass, a trait that in turn depended on growth form and dispersal syndrome. The dominant seedling type in the community was phanerocotylar epigeal with foliaceous cotyledons (56%), which was mostly associated with small seed mass and herbaceous growth form. Our results provide evidence that several plant life-history traits explain an important part of the variation in seed germination and seedling characteristics observed among species. Therefore, these plant life-history traits may be useful for grouping species in terms of their establishment strategies and roles on the regeneration of tropical dry communities.

**Funding:** The first author thanks the Posgrado en Ciencias Biologicas of Universidad Nacional Autónoma de México (UNAM) for its support during his PhD studies. This work was funded by the Programa de Apoyo a Proyectos de Investigación e Innovación Tecnológica (PAPIIT) of UNAM, Project IN207512 (Fenología de especies arbóreas del bosque tropical caducifolio en la Depresión del Balsas, Michoacán) (GIM as Head Researcher). The funders had no role in study design, data collection and analysis, decision to publish, or preparation of the manuscript.

**Competing interests:** The authors have declared that no competing interests exist.

# Introduction

The seed-to-seedling transition is one of the most vulnerable phases in the life cycle of a plant [1,2]. In seasonally dry ecosystems, successful seedling establishment largely depends on the seed resources and rapid acquisition of available environmental resources during a short time; thus, seed germination and seedling growth are expected to occur rapidly during the wet season [3,4]. Nonetheless, different seed germination patterns and functional seedling strategies co-occur in these ecosystems. Germination and seedling characteristics are likely correlated with other plant life-history traits (e.g., growth form, phenology, seed mass), which jointly could be influencing species' germination and their establishment [5]. For example, seed mass can be positively related to the germination rate of herbaceous species [6]. Moreover, larger seeds have also been associated with larger seedlings, which may, in turn, have a higher survival rate during establishment, compared to seedlings originating from smaller seeds [7,8].

Seed dispersal phenology can allow seeds to temporarily escape conditions that are unfavorable for seed germination and seedling establishment [9]. In ecosystems with strong seasonality in water availability, seeds of herbaceous plants are expected to disperse at the end of the rainy season [10]. These seeds commonly have some type of dormancy mechanism (e.g., physiological or physical) that allows them to delay germination until the end of the dry season or the beginning of the rainy season [11]. Germination patterns have also been associated with dispersal syndromes. For instance, some species have seeds dispersed by animals with a hard and impermeable seed coat and require mechanical or chemical scarification to germinate, while wind-dispersed seeds frequently lack physical dormancy and germinate rapidly [12].

The relationship between seed germination and different life-history traits has been examined at the species level, and generally for a single trait, rather than considering multiple traits at the community level [13,14]. Furthermore, many plant traits are not phylogenetically independent, and some ecological relationships can be the result of the species evolutionary history [15]. For example, seed mass is a phylogenetically conserved trait at the family and genus levels in different communities [16,17] and is related to seed dispersal (e.g. positive association between seed mass and biotic dispersal), and growth form (e.g. positive association between seed mass and plant height), as well as to the emergence rates and establishment of seedlings [18]. Variations in seed germination at the community level, considering the phylogeny and plant life-history traits, have been studied in seasonal communities [10,13,19], but this approach is underrepresented in tropical dry forests.

Also, little explored for tropical dry forests are interspecific differences in seedling establishment strategies (but see [20,21]), and our knowledge on how seedling characteristics relate to other plant life-history traits is very limited. In seedlings, the transition from dependence on seed reserves to dependence on external resources can be inferred from the cotyledons' functional morphology [22]. Characterizing seedling types according to the position, exposure, and function of the cotyledons, can reveal important insights about seedling establishment strategies [23, 24]. For example, in tropical dry forests, seedlings with photosynthetic cotyledons are the most frequent, and they begin to depend on light and photosynthesis earlier than those with reserve cotyledons [25].

Variation in seed germination patterns and seedling characteristics, analyzed at the community level, can provide useful information to improve our understanding of regeneration dynamics and species coexistence in tropical dry forests [26], however, this information, when available, comes mainly from studies on tree species, that have life-history strategies different from other growth forms [9,27,28]. Thus, the information known until now only shows a fraction of all the variation in germination and seedling establishment, and furthermore of all the

potential establishment strategies that can occur in tropical dry forests. Here, we studied the variation in seed germination patterns and seedling types at the community level, in association other plant life-history traits and the phylogenetic relatedness among species of a Neotropical dry forest. First, we characterized life-history traits that can be important on plant establishment, such as the dispersal phenology, dispersal syndromes, growth forms, and seed mass. Then we try to answer the following questions: to what extent the interspecific variation in seed germination and seedling types can be explained by plant life-history traits and phylogeny? and what is the relationship between seedling types and germination patterns?

Given the strong seasonality of water availability in our study site, we predicted that seed germination would be more frequent in species that disperse their seeds at the end of the dry or beginning of the rainy season. We also expected that germination patterns (lag time, rate and percentage) would be correlated with seed mass, but that these relationships would depend on growth form and dispersal syndrome. Particularly, we expected that in herbaceous species the lag time and rate of germination would be faster in seeds with smaller mass, compared to heavier seeds. Particularly, we expected that in herbaceous species the lag time and rate of germination would be faster in smaller seeds, compared to heavier seeds. Since lag time and rate of germination can vary with the thickness of the seed coat [11], we expected that germination to be faster in seeds with a thin coat such as wind-dispersed species in comparation with seeds with a thick coat such as animals-dispersed species. Due to the fact that functional morphology of cotyledons (seedling types) is important for seedling establishment, we expected that seedling types are associated with other plant life-history traits. Particularly, we predicted that seedlings with foliaceous cotyledons would occur more frequent in herbs (they have predominantly small seeds, therefore the lag time and germination rate would be faster), whereas seedlings with reserve cotyledons would occur more frequent in woody species (associated more often with large seeds, therefore the lag time and germination rate would be slower).

## Material and methods

### Study area

This study was conducted in the Municipality of Churumuco, Michoacán, Mexico (18˚38'-44' N; 101˚38'-41' W; ca. 300 m a.s.l.). The climate in the study area is semiarid and warm (BSh), with a total annual precipitation of 564 mm [29]. Precipitation is seasonal, with rainfall occurring between June and September, followed by a long dry season. The annual mean temperature is 29.4˚C, ranging from 23 to 36˚C throughout the year (Fig 1). The predominant vegetation is the tropical dry forest, mostly secondary forests with more than 50 years of abandonment after agriculture. Common arboreal species include *Backebergia militaris*, *Bursera crenata*, *Bursera infernidialis*, *Bursera sarukhanii*, *Cordia elaeagnoides*, *Erythrina oliviae*, *Gossypium lobatum*, *Handroanthus impetiginosus*, *Heteroflorum sclerocarpum*, *Lonchocarpus huetamoensis*, *Manihot crassipetala*, *Pachycereus tepamo*, *Poincianella eriostachys*, *Senegalia picachensis*, *Stenocereus fricii* and *Stenocereus quevedonis*. Shrub species, which include *Lantana hirta*, *Pouzolzia guatemalana*, *Randia thurberi*, *Salpianthus standleyi*, *Senna obtusifolia* and *Zapoteca formosa* constitute another important component of this forest, together with liana species such as *Combretum fruticosum*, *Galactia acapulcensis*, *Ipomoea robinsonii*, *Ipomoea suaveolens* and *Operculina pteripes*. Examples of common herb species include *Aldama michoacana*, *Cnidoscolus calyculatus*, *Euphorbia graminea*, *Mammillaria beneckei*, *Melinis repens*, *Salvia uruapana*, *Senna uniflora*, *Tagetes erecta* and *Zinnia flavicoma*.

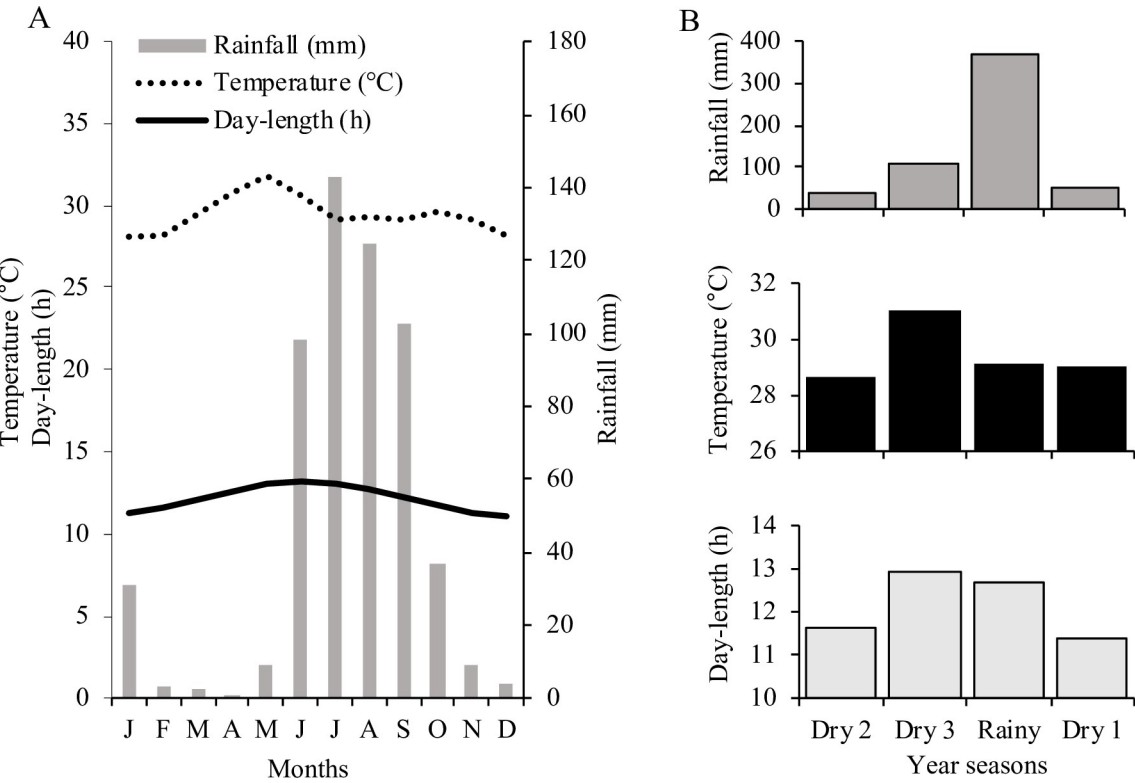

**Fig 1.** (A) Monthly data for precipitation, mean temperature and day length over a period of 30 years (1980–2010) and (B) the four seasons recognized in the study area: Dry 1 (from 21-Sept to 20-Dec), Dry 2 (from 21 Dec to 20-Mar), Dry 3 (from 21-Mar to 20-June) and Rainy (from 21-June to 20-Sept). Climatic data were obtained from the meteorological station of Churumuco (at latitude 19˚), Michoacán, Mexico, approximately 10 km from our study site.

### Plant life-history traits

**Seed dispersal phenology.**   Based on monthly fruit phenology observations (January 2013-December 2014), we characterized the timing of seed dispersal (the time period during which mature fruits were observed). Phenological observations were made directly on all plant species occurring along of a 6 km long and 15 m wide transect. We recorded data for 110 species: 53 trees, 16 shrubs, 12 lianas, 5 herbaceous climbers and 24 herbs (S1 Table). Using circular statistics, we calculated the mean angle (ā) of seed dispersal phenology, which indicates the mean date of seed dispersal [30]. To calculate ā, months were converted into angles (e.g., January corresponds to 15˚, February to 45˚, and so [30], and the statistical significance of ā was determined using Rayleigh's test (z). By taking into account the changes in rainfall, temperature and day-length throughout the year in the study site, we classified species with mature fruit as: (i) rainy season species, from 21-June to 20-September; (ii) onset-of-dry-season species (Dry 1), from 21-September to 20-December; (iii) middle-of-dry-season species (Dry 2), from 21 December to 20-March; and, (iv) late-dry-season species (Dry 3), from 21-March to 20-June (Fig 1).

**Dispersal syndromes and seed mass.**   Seed dispersal syndromes were determined based on the morphology of the diaspores and field observations. Following van der Pijl [31], we classified species as: (i) autochorous, including ballochorous (ballistic dispersal) and barochorous (dispersal through gravity) diaspores, (ii) anemochorous, including pterochorous (winged diaspores) and pogonochorous (plumed diaspores) diaspores, and (iii) zoochorous species,

including endozoochorous (internal dispersal by vertebrates) and epizoochorous (dispersal by adhesion to the external parts of animals) diaspores. We collected 10 seeds from 10 individuals of each species (n = 100) to obtain the fresh seed mass using an analytical balance (Ohaus-Explorer®, E11140, Pine Brook, NJ, USA), with a precision of 0.01 mg.

**Seed germination.** For each species, we collected 10 mature fruits that showed no physical damage or apparent disease, from five individuals (n = 50). We extracted the seeds from the fruits and then separated viable from non-viable seeds using the flotation technique, except in the case of species with very lightweight seeds [32]. In the latter case, seeds were squeezed lightly with a fine tweezer to select the firmest ones; soft seeds were considered to be non-viable. Seed germination experiments were performed at the study site in a shade house with 80% shade cover. We used black polythene bags (20 × 15 × 30 cm) filled with local soil collected from the same location (sand and clay in a proportion 60:40, respectively), and since one of the objectives was to germinate the seeds in the most similar conditions which occurs in their habitat, we did not apply any pesticides or fertilizers to the soil. Seeds were stored in paper bags and sown 1 or 2 days after they were collected. For each of the five parental trees, we placed 10 seeds per bag (50 seeds per species). Each seed was buried at a depth of approximately twice the seed width to prevent desiccation of seed and to facilitate its emergence; we applied water every third day with a manual sprinkler. Radicle emergence (germination) was recorded every third day for a maximum period of 60 days. The accumulated germination data was fitted to an exponential sigmoid curve using the program Table Curve 2D, version 3 (AISN, Software, Chicago, IL, USA), and we calculated the lag time of germination and the germination rate from the slope at the inflection point of the curve (maximum first derivative).

**Seedling types.** For species where germination did not occur, a pregerminative treatment (e.g., physical or chemical scarification) was applied to a new set of seeds to induce germination and obtain seedlings. We used only the data from non-scarified seeds for germination analysis. When seedlings had 2–3 leaves, we collected five seedlings of each species to determine their functional morphology. Following Garwood [23], who identified eight seedling types based on the position (epigeal or hypogeal), exposure (phanerocotylar or cryptocotylar) and function (foliaceous or reserve) of the cotyledons, we found three types of seedlings: (i) PEF, phanerocotylar epigeal with foliaceous cotyledons, (ii) PER, phanerocotylar epigeal with reserve cotyledons and (iii) CHR, cryptocotylar hypogeal with reserve cotyledons.

**Phylogenetic tree.** We constructed a phylogenetic tree for 110 species using the Phylomatic command of the software PHYLOCOM [33]. Phylogenetic reconstruction was conducted based on Angiosperm Phylogeny Group (tree: R20120829) [34]. Even though the tree was resolved to the family level, it was not possible to resolve the existing polytomies due to the scarcity of molecular data and phylogenetic studies on the species in the study area. We used the bladj algorithm in the software PHYLOCOM [35] to calibrate the length of the phylogenetic branches based on the ages of divergence of the largest angiosperm clades that were proposed by Magallón et al. [36].

## Data analysis

Using the Chi-squared goodness of fit test, the frequency of species with different dispersal phenology was evaluated. We test the relationships between the germination response variables (lag time, rate and percentage of germination) and the explanatory variables (seed mass, seed dispersal phenology, growth form, dispersal syndrome), with a phylogenetic generalized least squares (PGLS) model in R [37] using the package 'caper' [38]. This method incorporates a phylogenetic covariance matrix of the studied species into the model and uses Pagel's lambda

(λ) as a measure of the phylogenetic signal, calculated through the maximum verisimilitude method. When λ = 0 there is phylogenetic independence, but when λ = 1 there is a strong phylogenetic signal [39]. Model selection was based on the Akaike information criterion (AIC; S2 Table) [40]. Since seed mass differed by various orders of magnitude among species, the natural logarithm of this variable was used for all subsequent analyses. The germination percentage data were arcsine transformed to be included in the analyzed models. Given the low frequency of herbaceous climbing species and their similarities to herbs in terms of the traits studied, both growth forms were combined into a single group. Therefore, growth forms considered in all analysis were trees, shrubs, lianas and herbs. The same process was applied to pterochorous and pogonochorous diaspores, which formed a single group (anemochorous species) and to ballochorous and barochorous plants, which were combined into the autochorous group.

Given the low frequency of cryptocotylar seedlings with reserve cotyledons, these were excluded from analysis. Germination of the studied species and their seedling type were codified as binary dependent variables and related to growth form, seed dispersal phenology and dispersal syndrome (predictor variables) [41]. The method "logistic IG10" was used in all models [42]. The degree of phylogenetic correlation was estimated through the parameter alpha (α). We ran 'phyloglm' with bootstrap of n = 1000 to obtain 95% confidence intervals for alpha and effects of parameters [41]. The established limits for α range from -4 (no phylogenetic signal is present) to 4 (presence of phylogenetic signal). The analysis was conducted using the package 'phylolm' [42] in R. We also used PGLS model to evaluate the seed germination patterns and seed mass in relation to seedling type. Finally, in order to visualize the patterns of plant life-history traits along the phylogenetic tree, we plotted the phylogenetic tree with the continuous (seed mass, lag time, rate and percentage of germination), and categorical traits (growth form, dispersal syndrome, dispersal phenology, and seedling type) using the packages 'ape', 'ggtree' and 'ggplot2' in R [43, 44].

## Results

### Plant life-history traits

Autochorous species were the most frequent in the community studied (ballochorous 9% and barochorous 40%), following of zoochorous (endozoochorous 28% and epizoochorous 6%) and anemochorous species (pogonochorous 6% and pterochorous 10%; $\chi^2$ = 63.34, d.f. = 5, P < 0.001). Ninety-two percent of the studied species dispersed their seeds during the dry season. The number of species varied significantly between the four periods of the year ($\chi^2$ = 60.54, d.f. = 3, P < 0.001). Fifty-six percent of the species dispersed their seeds at the beginning of the dry season (dry 1; October-December), 19% did so in the middle of the dry season (dry 2; January-March) and 17% at the end (dry 3; April-June). Only 8% of the species dispersed their seeds during the rainy season (July-September). On average, the weight of the seeds of all species under study was 0.126 ± 0.03 g, ranging from 0.0010 g (*Otopappus epaleaceus*, Asteraceae) to 2.34 g (*Spondias purpurea*, Anacardiaceae) (S2 Table and Fig 2).

### Seed germination

We observed germination in 74% of the species, which were associated with seed dispersal phenology and growth form (Table 1). The percentage of germination in the plant community ranged from 0% (29 species) to 100% (3) and was > 50% in 34 species (Fig 2). No relationship was found between any plant trait and the percentage of germination ($R^2$ = 0.10, P = 0.63, λ = 0.00).

The model that included growth form in interaction with seed mass, and dispersal syndrome explained most of the variation in germination lag time, with no phylogenetic

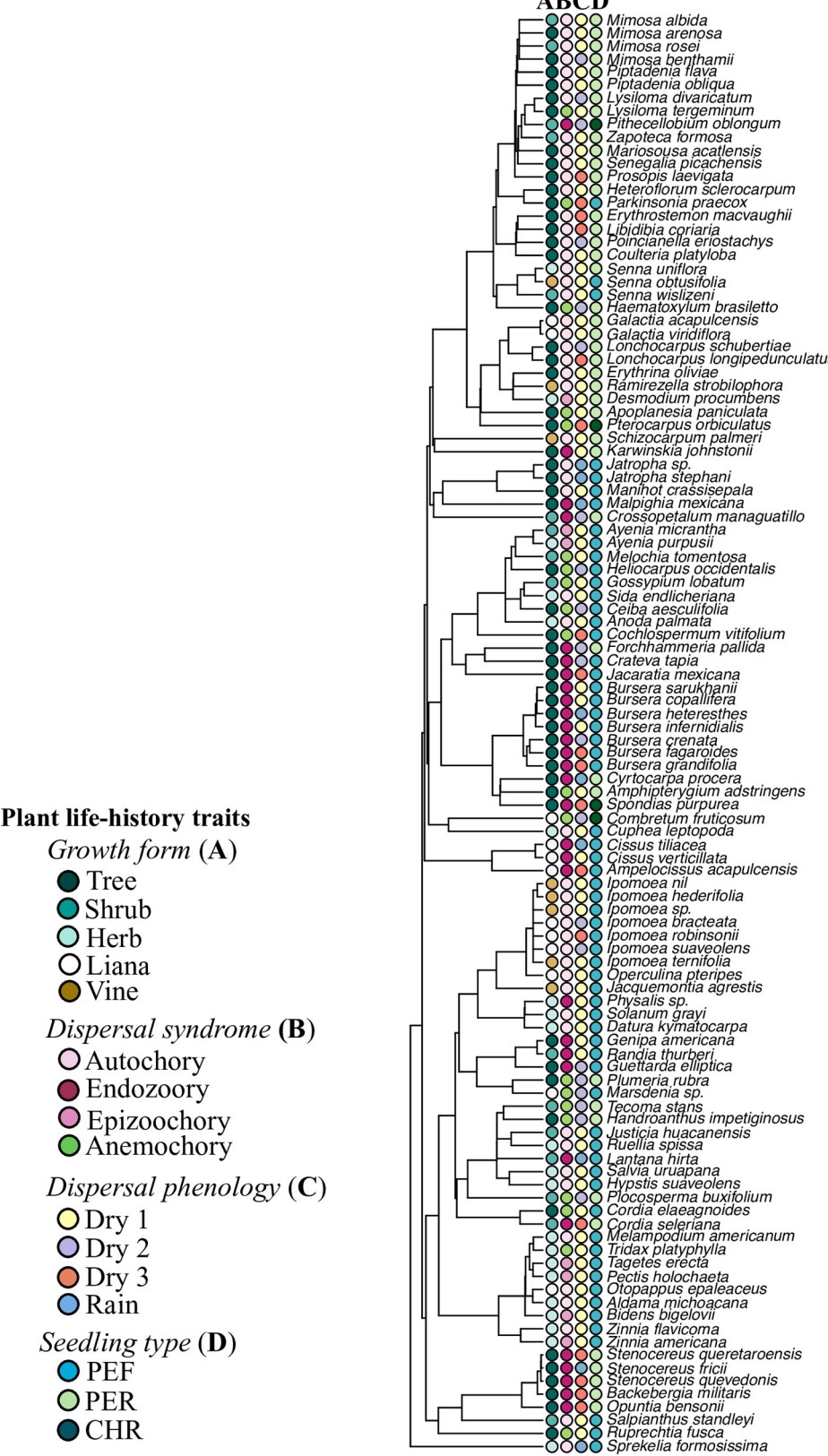

**Fig 2. Distribution of plant life-history traits at the tips of the phylogenetic tree constructed for 110 species of a tropical dry forest.**

**Table 1. Results of the phylogenetic generalized least squares models used to evaluate the relationships of germination lag time and germination rate (response variables) with growth form, dispersal syndrome, seed dispersal phenology and seed mass (SM).**

| Model and traits | Estimate | t-value | P-value |
|---|---|---|---|
| Lag time ~ dispersal syndrome + growth form / seed mass | | | |
| Intercept | 2.41 | 12.64 | < 0.001 |
| Autochory | 0.39 | 2.41 | 0.01 |
| Endozoochory | 0.21 | 1.25 | 0.21 |
| Epizoochory | 0.6 | 1.85 | 0.06 |
| Herb/SM | 0.17 | 4.01 | < 0.001 |
| Liana/SM | 0.1 | 1.72 | 0.08 |
| Shrub/SM | 0.12 | 2.5 | 0.01 |
| Tree/SM | 0.03 | 0.65 | 0.51 |
| Rate ~ dispersal syndrome / seed mass + growth form / seed mass | | | |
| Intercept | 30.05 | 7.23 | <0.001 |
| Autochory/SM | 3.81 | 2.78 | 0.006 |
| Endozoochory/SM | 2.55 | 1.84 | 0.06 |
| Epizoochory/SM | 6.08 | 3.27 | 0.001 |
| Herb/SM | -5.12 | -2.88 | 0.005 |
| Liana/SM | -4.54 | -2.24 | 0.02 |
| Shrub/SM | -3.69 | -2.13 | 0.03 |
| Tree/SM | -1.04 | -0.63 | 0.52 |

The forward slash indicates the interaction between traits.

relationship ($R^2$ = 0.24, $P$ = 0.006, $\lambda$ = 0.00; Table 1, S1 Fig). On average, seed germination (lag time) at the community level began after 12.06 ± 5.84 days. Germination occurred within a shorter time period for herb species (8.66 ± 4.92 days), and within a longer period in the case of tree and liana species (13.48 ± 5.89 and 12.49 ± 5.45 days, respectively). Autochorous species were also associated with longer lag times (12.54 ± 6.9 days), while the epizoochorous species showed shorter lag times (8.86 ± 0.62 days) than the species with other dispersal syndromes (Fig 3).

Mean germination rate was 30.32 ± 15.08% day$^{-1}$ and was also associated with growth form, dispersal syndrome, both traits in interaction with seed mass ($R^2$ = 0.18, $P$ = 0.02, $\lambda$ = 0.00; Table 1, S1 Fig). Herbs and epizoochorous species, which had lightweight seeds, showed higher rates of germination (38.07 ± 11.72 and 35.61 ± 12.92% days$^{-1}$, respectively; Fig 3) than tree and endozoochorous species (26.98 ± 12.59 and 28.43 ± 14.23% day$^{-1}$, respectively; Fig 3).

## Seedling types

The phanerocotylar epigeal with foliaceous cotyledons (PEF) seedling type was the most common in the plant community under study (56% of the species), followed by phanerocotylar epigeal with reserve cotyledons (PER, 39%) and cryptocotylar hypogeal with reserve cotyledons (CHR, 5%; $\chi^2$ = 16.56, d.f. = 2, $P$ = 0.002). Trees had most frequently PER seedlings, while PEF seedlings were associated with herb and liana species (Table 2). Seedling type was independent of dispersal phenology and dispersal syndromes (Fig 2 and Table 2).

Seedling type showed a significant relationship with seed mass, with the presence of a strong phylogenetic signal ($R^2$ = 0.10 $P$ = 0.01, $\lambda$ = 0.63). Species with CHR seedlings were associated with the heaviest seeds (0.56 ± 0.82 g), followed by species with PER seedlings (0.12 ± 0.26 g), while those that had PEF seedlings came from lighter seeds (0.09 ± 0.31 g).

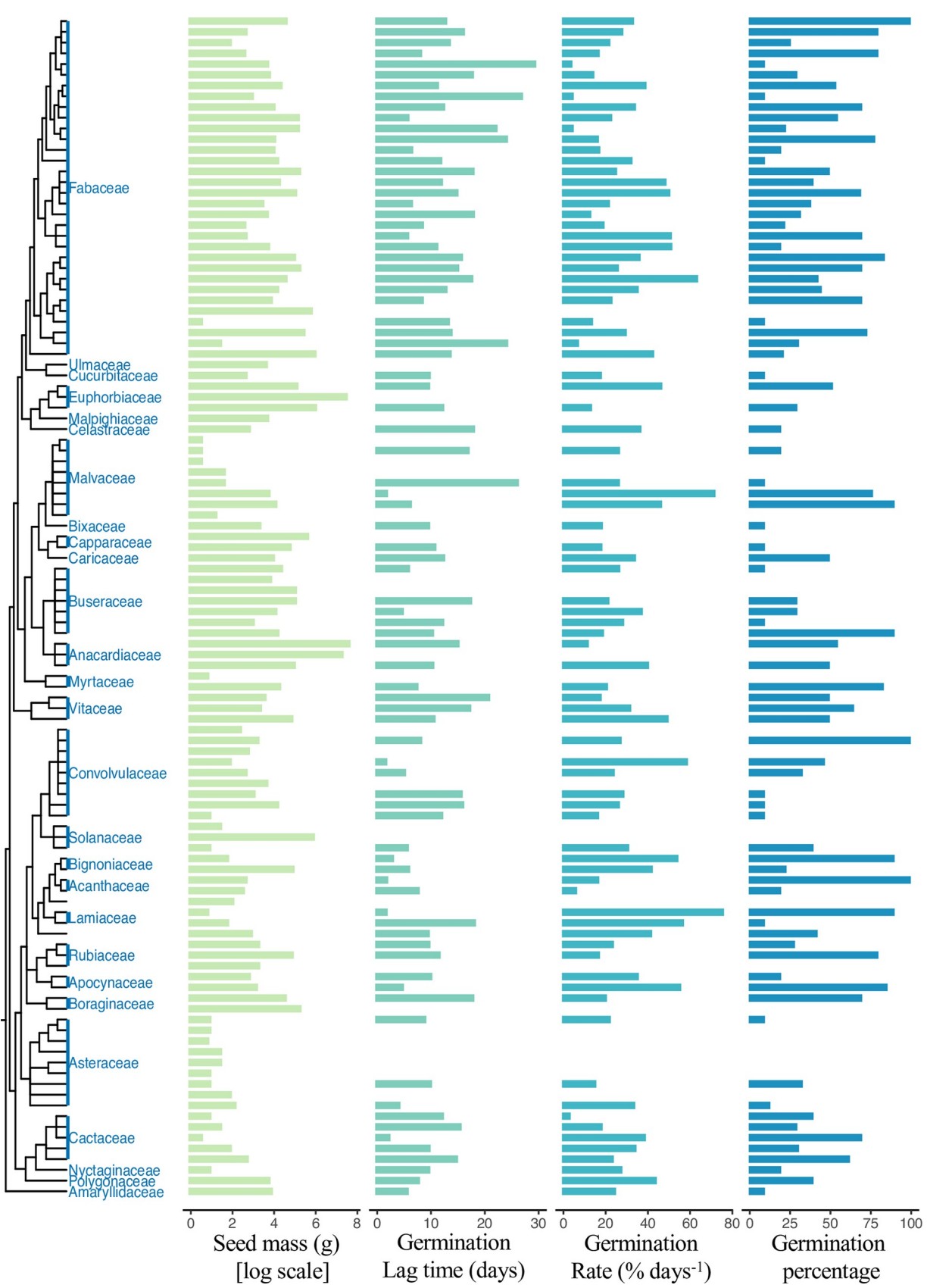

**Fig 3. Seed mass and seed germination (rate, lag time and percentage of germination) according to the structure of the phylogenetic tree built from 110 plant species in a tropical dry forest of Mexico.**

## Seed germination patterns of seedling types

Germination showed no significant relationship with seedling type. Lag time and germination rate were related with seedling type, with no phylogenetic influence ($R^2 = 0.16$, $P < 0.001$, $\lambda = 0.00$; $R^2 = 0.07$, $P = 0.01$, $\lambda = 0.0$, respectively). For species with PER seedlings, lag time was

**Table 2. Results of logistic phylogenetic regression carried out to analyze the relationships of germination and seedling types, with different functional attributes.**

| Parameter | Estimate | S.E. | z-value | Bootstrap mean | Bootstrap 95% IC | P-value |
|---|---|---|---|---|---|---|
| Germination ~ Growth form + Dispersal syndrome + Dispersal phenology | | | | | | |
| Alpha | 0.003 | | | 0.002 | 0.0019, 0.003 | |
| Intercept | 1.033 | 0.93 | -1.1 | -1.59 | -1.1, -1.59 | 0.27 |
| Lianas | 1.77 | 0.94 | 1.88 | 1.57 | 1.88, 1.57 | 0.06 |
| Shrubs | 1.86 | 0.8 | 2.31 | 1.84 | 2.31, 1.84 | 0.02 |
| Trees | 1.59 | 0.74 | 2.15 | 1.41 | 2.15, 1.41 | 0.03 |
| Autochory | 0.97 | 0.69 | 1.39 | 0.89 | 1.39, 0.89 | 0.16 |
| Endozoochory | -0.27 | 0.68 | -0.39 | -0.13 | -0.39, -0.13 | 0.68 |
| Epizoochory | 0.28 | 1.07 | 0.26 | 0.19 | 0.26, 0.19 | 0.78 |
| Dry 2 | 1.53 | 0.87 | 1.75 | 1.45 | 1.75, 1.45 | 0.07 |
| Dry 3 | 0.51 | 0.61 | 0.83 | 0.27 | 0.83, 0.27 | 0.4 |
| Rain | -1.96 | 0.98 | -1.99 | -1.82 | -1.99, -1.82 | 0.04 |
| PER ~ Growth form + Dispersal syndrome + Dispersal phenology | | | | | | |
| Alpha | 0.003 | | | 0.002 | | |
| Intercept | -1.03 | 1.05 | -1.1 | -2.64 | 1.05, -2.64 | 0.27 |
| Lianas | 1.77 | 0.9 | 1.88 | 0.33 | 0.9, 0.33 | 0.06 |
| Shrubs | 1.86 | 0.77 | 2.31 | 1.65 | 0.77, 1.65 | 0.02 |
| Trees | 1.59 | 0.77 | 2.15 | 2.57 | 0.77, 2.57 | 0.03 |
| Autochory | 0.97 | 0.65 | 1.39 | 1.43 | 0.65, 1.43 | 0.16 |
| Endozoochory | -0.27 | 0.65 | -0.39 | -0.27 | 0.65, -0.27 | 0.68 |
| Epizoochory | 0.28 | 1.24 | 0.26 | 0.73 | 1.24, 0.73 | 0.78 |
| Dry 2 | 1.53 | 0.58 | 1.75 | -0.01 | 0.58, -0.01 | 0.07 |
| Dry 3 | 0.51 | 0.56 | 0.83 | 0.37 | 0.56, 0.37 | 0.44 |
| Rain | -1.96 | 1.02 | -1.99 | -0.83 | 1.02, -0.83 | 0.04 |
| PEF ~ Growth form + Dispersal syndrome + Dispersal phenology | | | | | | |
| Alpha | 0.0019 | | | 0.002 | 0.0019, 0.003 | |
| Intercept | 1.84 | 1.07 | 1.71 | 1.97 | 1.23, 2.44 | 0.08 |
| Lianas | -0.59 | 0.78 | -0.75 | -0.81 | -1.09, -0.25 | 0.44 |
| Shrubs | -1.79 | 0.68 | -2.61 | -1.93 | -2.27, -1.48 | 0.009 |
| Trees | -1.25 | 0.79 | -1.76 | -1.37 | -2.09, -0.72 | 0.07 |
| Autochory | -0.4 | 0.55 | -0.73 | -0.43 | -0.93, -0.32 | 0.45 |
| Endozoochory | -0.1 | 0.57 | -0.18 | -0.18 | -0.5, 0.42 | 0.85 |
| Epizoochory | -0.21 | 1.06 | -0.19 | -0.29 | -0.64, 0.16 | 0.84 |
| Dry 2 | -0.39 | 0.47 | -0.82 | -0.33 | -0.93, 0.11 | 0.4 |
| Dry 3 | -1.12 | 0.59 | -1.91 | -0.86 | -1.19, -0.62 | 0.05 |
| Rain | 0.09 | 0.65 | 0.14 | 0.27 | -0.37, 0.64 | 0.88 |

Abbreviations: PEF (phanerocotylar epigeal with foliaceous cotyledons), PER (phanerocotylar epigeal with reserve cotyledons) and S.E.: standard error.

longer (12.93 ± 7.19 days) compared to that observed in the species with PEF seedlings
(6.57 ± 6.75 days). Species with PER seedling types also showed high rates of germination
(26.33 ± 16.78% day$^{-1}$) compared to the PEF seedlings (19.92 ± 20.17% day$^{-1}$).

## Discussion

### Plant life-history traits

In the tropical dry forest studied, only 8% of the plant species dispersed their seeds during the
rainy season. This phenological pattern contrasts with other studies carried out in seasonally
dry ecosystems, in which a high percentage of species disperse during rainy season (85% [45],
42% [46], 33% [47]). The dispersal of seeds during the dry season in this community may be
associated with the life cycle of the herbs. Herb species bloom at the end of the rainy season, so
seed dispersal occurs at the beginning of the dry season, and most of these species have dry dia-
spores dispersed by autochory [48]. In woody species both blooming and seed dispersal
occurred during the dry season [48,49]. The observed dispersal phenology in our study com-
munity may suggest that the seeds of most plants may have some mechanism of drought toler-
ance (e.g. orthodox seeds, quiescence, physic dormancy), given that 7–8 months must elapse
until the following rainy season. However, we did not assess the mechanism of drought toler-
ance for the studied seeds; although this is an important topic, relevant for the dynamics of
vegetation regeneration. Its evaluation must be given priority particularly considering the cli-
mate change scenarios, which may impose more stronger drought conditions that may affect
species differently, depending on the mechanisms that their seeds have to deal with this
adverse condition.

### Seed germination and seedling types

At the community level, we found different seed germination patterns and in most of the spe-
cies asynchronous germination occurred, i.e., not all seeds germinated at the same time. This
is an interesting pattern that could be considered as an adaptive strategy to cope with temporal
unpredictable of rainfall. In this way, in species that germinate in a staggered manner, it is pos-
sible that some event of germination will coincide with the optimum time for seedling estab-
lishment. Our results suggest that seed dispersal phenology has an important influence on seed
germination (occurrence or nonoccurrence of germination). Specifically, germination did not
occur in 70% of species that disperse seeds at the beginning of a dry season. As has been sug-
gested by other studies, in plants dispersing their seeds during the dry season, mechanisms
such as dormancy could play an important role for the occurrence of germination at the opti-
mal time for the successful establishment of seedlings in seasonally dry tropical forests [12,46].
Germination was positively linked to trees, but negatively associated with the herbs, which dis-
persed their seeds at the beginning of dry season. Absence of germination in herb species has
been observed in other studies and it has been proposed that this may be a strategy to escape
unfavorable conditions [50]. Many seeds of annual species remain dormant and become part
of the seed bank in what has been interpreted as a strategy for ensuring successful seedling
establishment in strongly seasonal environments. The presence of herbs in the seed bank may
have a strong impact on the regeneration dynamics of tropical dry forests since germination of
these seeds during the rainy season could limit the establishment of other species. For example,
intraspecific and interspecific competition among seedlings could increase when a high per-
centage of herb seeds germinate. However, subsequent studies should assess the impact of
these interactions on species establishment, especially during the beginning of the rainy sea-
son, when seedling density is higher.

The association of seed mass with growth form was significantly related to the lag time of germination. Herbs, which had on average have lighter seeds, germinated in less time compared to trees and lianas. The relationship between growth form and germination lag time has been documented in other studies conducted in temperate and alpine communities [11,19]. It is generally considered that shorter times to initiation of germination confer competitive advantages to species with small seeds [9,50]. Trees and autochorous species, which showed a delayed germination response, had the lowest rates of germination. Likewise, the herb species, and those that disperse through epizoochory, showed short lag times and higher germination rates. These germination patterns in tropical dry forest species could be related to the duration of the plant's life cycle. In herb species with short life cycles, a rapid germination response can increase the probability of seedling establishment and could also allow the plants to reach a larger size before reproduction [51]. The slow germination observed in the woody species could be related to the presence of larger seeds that contain more reserves for seedling establishment [19,52]. In agreement with the studies by Grimme et al. [10] and Xu et al. [19], epizoochorous species were associated with a rapid germination, while the autochorous plants showed a more delayed response. It has been proposed that species with a lower dispersal capacity (such as autochorous species) also have delayed germination, and that the species associated with mechanisms that favor dispersal over greater distances have a more rapid germination [11,13]. The germination patterns documented at the community level could be related to distinct establishment strategies, that have possibly differential impacts on the regeneration dynamics of tropical dry forests. However, it is necessary to evaluate these patterns in long-term studies, including other functional trait not considered here (e.g. seedling root and stem traits) and abiotic factors (e.g. soil characteristics, microclimatic conditions) that could determinate plant establishment.

The frequency of different seedling types in a plant community has been related to different plant functional traits that can influence seedling establishment [8,24]. Our results support the proposal that phanerocotylar epigeal with foliaceous cotyledons seedlings (PEF) are more common in Neotropical forests [23,24]. Seedlings of type PEF have a greater initial allocation of photosynthetic tissue, which could be important in sites such as dry forests, where seedlings should take advantage of the period in which the rainy season begins, and when the availability of light is not yet a limiting factor [53,54,55].

Species with cryptocotylar hypogeal with reserve cotyledons (CHR) and phanerocotylar epigeal with reserve cotyledons (PER) seedlings had heavier seeds than those with PEF seedlings. This result is consistent with the proposal that larger seeds are associated with hypogeal cotyledons, while the trend for smaller seeds to have epigeal and foliaceous cotyledons has been documented elsewhere [8,23,24]. However, it is interesting to note that the phylogeny of the species had an important effect on the correlation between seed mass and seedling type. This is consistent with the notion that seed mass is a trait that is conserved throughout different taxonomic levels, such as order, family and genus [17,53,56,57]. Different studies suggest that seed mass and cotyledon types show correlated evolutionary changes [8,24,55]. Seed mass affects the capacity for dispersal and the probability of seedling establishment; for this reason, it is a highly important ecological characteristic and is considered an evolutionarily stable trait of plant species [58,59]. For instance, we found an important phylogenetic conservatism in seedling types within the Asteraceae (PEF), Burseraceae (PEF), Cactaceae (PER) and Convolvulaceae (PEF) (Fig 2). These families are significant components of tropical dry forest, related to different successional stages and soil characteristics. For example, a large number of species of the Asteraceae and Convolvulaceae are highly dominant in the early successional stages, while the species of the Burseraceae and Cactaceae are successful elements in rocky soils, generally in later successional phases (Cortés-Flores, unpublished data).

Herb species had mostly PEF seedlings and showed shorter germination lag time, whereas woody species were mainly related to PER seedlings and longer germination lag times. These relationships may suggest different plant strategies related to seedling establishment. For example, in woody species, the presence of reserve cotyledons associated with larger seeds may give these plants a greater probability of survival, since the additional metabolic reserves in large seeds could compensate for the loss of carbon that can occur due to unpredictable rainfall events [25,54] or episodes of herbivory, or allow for the development of deep roots that facilitate the absorption of soil water. In herb species, prompt germination and the presence of seedlings with photosynthetic cotyledons allows for a rapid establishment [8,60]. The strategy observed in herb species could allow them to efficiently use temporarily available water, since the rainy season in the tropical dry forests is characterized by episodes of precipitation followed by periods of drought. However, a change in the distribution of precipitation is currently occurring due to global climate change, and rain episodes are now more frequently followed by prolonged periods of drought [61]. In this context, we observed high mortality in some herb species (e.g. *Salvia uruapana*, *Senna obtusifolia*, *S. uniflora*) that responded to these episodes of rain. In this sense, it is necessary to develop long-term studies to assess the seed-to-seedling transition stages of dry-forest plant species, *in situ*, in order to predict their responses to current and future climate-change scenarios.

## Conclusions

Our results provide evidence that growth form, seedling types and other plant life-history traits such as seed mass and dispersal syndrome explain an important part of the variability in seed germination and seedling characteristics observed among tropical dry forest plant species. Therefore, these plant life-history traits may be useful for detecting groups of species that could have different establishment strategies and roles in the regeneration of tropical dry forest communities, which can provide valuable information to define restoration strategies. However, further studies are needed to evaluate seed germination and seedling development *in situ* during the regeneration of tropical dry forests and determine how the interspecific variation in seed germination and seedling characteristics at the community level can promote species coexistence in this ecosystem.

## Supporting information

**S1 Table. Growth form of species in a tropical dry forest, in the Municipality of Churumuco, Michoacán, Mexico.**
(DOCX)

**S2 Table. Selection of the phylogenetic generalized least squares models constructed to evaluate the correlations between attributes, based on the Akaike information criterion.**
GF: growth form, DS: dispersal syndrome, DP: seed dispersal phenology, SM: seed mass, AIC: Akaike information criterion, Δi: delta Akaike.
(DOCX)

**S1 Fig.** Interaction of growth form and dispersal syndrome with seed mass to explain the variation in the lag time germination (A), and the germination rate (B).
(PDF)

## Acknowledgments

We are indebted to the inhabitants of Ejido Llano de Ojo de Agua, in Churumuco, Michoacán, Mexico, who gave us support at different phases of the research and allowed us to work in

their land; we particularly thank Lorenzo Sánchez Sanchez and Misael Rojas López. We appreciate the critical and valuable review provided by J. Aaron Hogan and four anonymous reviewers. English translation was reviewed by Rosamond Coates.

## Author Contributions

**Conceptualization:** Jorge Cortés-Flores, Guillermo Ibarra-Manríquez.

**Data curation:** Jorge Cortés-Flores, Guadalupe Cornejo-Tenorio, María Esther Sánchez-Coronado, Alma Orozco-Segovia, Guillermo Ibarra-Manríquez.

**Formal analysis:** Jorge Cortés-Flores, Guadalupe Cornejo-Tenorio, María Esther Sánchez-Coronado, Alma Orozco-Segovia, Guillermo Ibarra-Manríquez.

**Funding acquisition:** Guillermo Ibarra-Manríquez.

**Investigation:** Jorge Cortés-Flores, Guadalupe Cornejo-Tenorio, María Esther Sánchez-Coronado, Alma Orozco-Segovia, Guillermo Ibarra-Manríquez.

**Methodology:** Jorge Cortés-Flores, María Esther Sánchez-Coronado, Guillermo Ibarra-Manríquez.

**Writing – original draft:** Jorge Cortés-Flores, Guadalupe Cornejo-Tenorio, Guillermo Ibarra-Manríquez.

**Writing – review & editing:** Jorge Cortés-Flores, Guadalupe Cornejo-Tenorio, María Esther Sánchez-Coronado, Alma Orozco-Segovia, Guillermo Ibarra-Manríquez.

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
