## [Decision Letter · Decision Letter 0]

26 Jul 2019

PONE-D-19-15449

Functional correlations of seed germination and seedling types in tropical dry forest species

PLOS ONE

Dear Guillermo Ibarra-Manríquez, Ph. D.

Thank you for submitting your manuscript to PLOS ONE. After careful consideration, we feel that it has merit but does not fully meet PLOS ONE’s publication criteria as it currently stands. Therefore, we invite you to submit a revised version of the manuscript that addresses the points raised during the review process.

Three reviewers have evaluated your manuscript in detail and have made numerous constructive contributions. While they find valuable information and are somewhat positive in their potential subject to improvement, they also have several main concerns on the presentation and content, so the manuscript requires major changes to meet our criteria for publication.

In general terms, I agree with the reviewers. In particular, they have doubts about the quality of the presentation in both structure and fluency; on the conceptual clarity of the introduction; in the analyzes -especially when considering phylogeny; and on the absence of hypotheses and clearly structured objectives to answer a general question that allows any sort of conceptual advance. They also made significant comments on the clarity of the discussion, regional scope and insufficient use of international published literature on the subject.

I think you need to solve these points before the manuscript could be considered suitable for publication in this journal. If you can work on the comments of the reviewers, we are looking forward to a new version.

Yours sincerely,

Pedro G. Blendinger, PhD

Academic Editor

PLOS ONE

We would appreciate receiving your revised manuscript by August 10, 2019. To enhance the reproducibility of your results, we recommend that if applicable you deposit your laboratory protocols in protocols.io, where a protocol can be assigned its own identifier (DOI) such that it can be cited independently in the future. For instructions see: http://journals.plos.org/plosone/s/submission-guidelines#loc-laboratory-protocols

We look forward to receiving your revised manuscript.

Kind regards,

Pedro G. Blendinger, PhD

Academic Editor

PLOS ONE

3. In your Methods section, please provide additional location information of the collection sites, including geographic coordinates for the data set if available.

Reviewers' comments:

Reviewer's Responses to Questions

**Comments to the Author**

1. Is the manuscript technically sound, and do the data support the conclusions?

Reviewer #1: Partly

Reviewer #2: Partly

Reviewer #3: Yes

2. Has the statistical analysis been performed appropriately and rigorously? 

Reviewer #1: No

Reviewer #2: Yes

Reviewer #3: Yes

3. Have the authors made all data underlying the findings in their manuscript fully available?

Reviewer #1: Yes

Reviewer #2: Yes

Reviewer #3: Yes

4. Is the manuscript presented in an intelligible fashion and written in standard English?

Reviewer #1: No

Reviewer #2: Yes

Reviewer #3: Yes

5. Review Comments to the Author

Reviewer #1: Review of PONE-D-19-15449

Guillermo Ibarra-Manríquez et al. “Functional correlations of seed germination and seedling types

Summary: Ibarra-Manríquez et al. seek to uncover the complex nature of species regeneration trade-offs of the plants in a tropical dry forest community in Michoacán Mexico. They employ a perspective that looks at the phenology of trees observed in the field, some greenhouse germination experiments and species characteristics (dispersal syndrome, seedling type, etc) to connect the dots and provide insight into the regeneration differences of and the variation plant strategies that are present that forest. Unfortunately, the manuscript lacks a sufficiently coherent organization or main message for me to recommend it for publication. Specifically, the questions are not as clearly formulated as they need to be and it is not clear how all of the pieces fit together to get at the questions. Overall, I found the paper difficult to read and understand. I found myself questioning about various details that would seem important to the study and in my opinion, the literature on the phenological and regeneration ecology of tropical dry is not referenced or incorporated well enough.

To that tune, I make several suggestions that aim to improve the manuscript:

Main comments:

1. Seedling type? The mention of seedling type does not appear until line 80, and then again on lines 90 and 92. In all three cases, it is introduced in passing. Seedling types are not explained until the methods starting on lines 173. If this is a key focus of the paper, where the authors aim to make a contribution, then both the formal definitions of each seedling type (with examples) and the logic behind their relationships to species phenology and regeneration dynamics need to be explained and emphasized from the beginning. When I read it at first (both on like 90 and 92) I was wondering what seedling type meant.

2. Phylogenetics – It is true, as the author state throughout, that phylogeny (i.e. plant lineage) is a predictor of functional trait values. For example, wood density and seed mass have phylogenetic structure. I do not think it is enough to simply point (e.g. L 69) out the fact that phylogeny matter, as that is common knowledge at this point. If you do discuss the effect of plant lineage, then you should state exactly how phylogeny structure the trait of interest. For example, higher order Angiosperms tend to have narrower root diameters than do lesser-evolved basal angiosperms and gymnosperms. You can move from that starting point to form some hypotheses about how things might be operating from there.

a. A second point about phylogenetics – In most of your analyses you seem to be looking for phylogenetic structure in the variables of interest. But this is not well justified. I could see many of these questions going the other way, as in, do relationships exist after accounting for phylogenetic structure in the data. For example is seed mass related to seedling type (after controlling for differences in seed mass due to plant lineage), which I think is what you are getting at. In that case, you would use phylogenetically independent contrasts, not pGLMs. Either way, you need to justify the use of the statistics and link them clearly back to questions that intended to answer and why and how they do so.

3. Hypotheses: you have none. Did you have some expectations when you started? List them. If you did not then at least state what the literature should propose (ie the starting point as mentioned above). The questions also run together throughout and are not clearly differentiable (See point 1)

4. “Functional traits?” only one of the things you call functional traits in your introduction is in fact a functional trait, based on the common definition of a function trait (Violle et al 2007 Oikos 116:882-892): ‘‘Functional traits’’ are defined as morpho-physio-phenological traits which impact fitness indirectly via their effects on growth, reproduction and survival, the three components of individual performance.” The one that is a true functional is seed mass (for reasons you more or less explain in your paper). The others are not. Phenology is phenology. The same goes for dispersal syndrome. Along the same lines (as I side not), I found the use of the terminology “germination response” cumbersome and overly complex. Was it not simple germination (whether you could grow the seedling or not)? Why not just use germination rates, and where nothing germinated the rate is zero? The germination rates and other demographic estimates are just that, indicators of species life-histories. They are not traits, they are not functional for anything. They are estimates for how the species perform (i.e germinates/ establishes) in the environment.

5. Models: Models need to be sufficiently explained and described using sufficient statistical detail to permit their re-creation from the data (which should be freely available). I did not find that to be the case in this work. Models were vaguely described (i.e no model formulae were given), and I found the result tables difficult to interpret. In many cases, it was unclear if multiple models were fit with multiple response variables or not. If multiple models are fitted, then each needs to be discussed as a separate entity, even if the co-variate structure of the models it the same.

6. Fitting the pieces together. / Organization. Have you ever heard anyone say “organization is half the battle?”…. well it’s true, in fact, it’s actually more than half the battle. It is imperative that information in scientific journals is organized in a fashion that permits its easy consumption (reading/referencing). There are conventions on how to lay out the figures and tables for a paper, similarly, there are conventions for the organization the flow of ideas and the results in the prose of the manuscript body. This work needs an overhaul/ massive re-organization of the information and structure of the writing. Subheadings were used, but do little to guide the reader through the cluttered mass of information. It is unclear which part of the study (the fieldwork vs. the greenhouse) the writers are referencing in many places. Also, concepts and ideas/ terminology need to be consistently used, defined from the beginning and organized in a manner that flows logically. Tables could be used in places where they are not (L116-127). And many of the main figures belong in the supplement and vice versa. The main findings of the pGLS models, which are the backbone of the study, appear in the supplement. I found these models impossible to reconstruct in my mind given the way they were presented.

Line comments:

L18: phases for natural regeneration. Is succession guided? If it is I have a hard time believing that it is guided by seed germination… I guess a case could be argued there but start with concrete clear sentences to draw the reader in, not tangential, controversial claims. Be cognizant of the words you choose.

L20: there is an important intraspecific variation? ?? check grammar and meaning

L 21: Delete, however; variation in seed germination

L22. Regeneration and succession

L 23: evaluated integrally?? Check Grammar

L24: response? Response to what?? Perhaps germination percentage, lag time and rate?

L25 I like dispersal method or mode better than syndrome.

L29: “we detected a strong phylogenetic signal” GIVE THE NUMBERS! Stats! Previous sentences in abstract say nothing of phylogenetics. How and why are you doing this? Need a preview.

L30: species that dispersed seeds

L31: germination was 30 percent compared to 50 percent for those that dispersed seeds at the end of the dry season

L 33: germination lag time.

L34: “showed an important relationship” what was the relationship??!

L36: Give number here. What is the trend? Big seeds from shade tolerant species have what type of seedling.

L41: are considered some of the most threatened ecosystem in the world. This seems like a weak ending to the abstract that is not at all related to the findings of the paper. What is your take-home message? Drive it home here, instead of this conservation pleading weak finish. Know your audience. Give them what they need to know.

L46: aka the seed to seedling transition (see the paper by Muscarella et al and references within)

L47: in seasonally-dry ecosystems? … seasonally-available water resources? (such as?)

L48; wet season?

L58: “is frequently found some innate” Grammar

L59: postpones?? Germination

L66: seed mass is the only functional trait here, the others are more life history characteristics or species demographic attributes.

L69: what is the phylogenetic trend of this trait?

L 74: see Lasky et al. Environment Research Letters. Wood Specific Gravity and taproot presence were shown to be important traits in a Puerto Rican dry forest.

L 75: This statement is not true! See Poorter & Markesteijn 2008/

L80: what is seedling type? Need definition

L83-85: This sentence is too vague

L86-89: so what? What is the point here? Why does it matter? Do you focus on more than just trees? That is not clear? Needs better logical flow

L93: move this up in the introduction. Get to this point first, which forms a solid jumping off point for this study.

L96 We predicted

L103: seed coat can vary with dispersal syndromes… how?

L94-107: These hypotheses are circular. Use the point about seed mass as a foundational starting point to flesh out questions about the seedling type and underlying demographics of them. It is not a good place to start your hypotheses, which I think are more about the other things, like how seedling type relates to seed mass and demographics, correct?

L117-127: list these in a table – give families and taxonomic authorities.

L121: Stenocereus – since when is cactus a tree??

Fig 1: Need to label y-axes. Where is the climate station in relation to the study site?

L136: how were the observations made? Using binoculars? Seed traps? These are very important details. They need to be included

L 139: why? What does this represent?

L146: Following the proposed classification by __________. Give name and year

L154: size? Mass is not size? Did you measure the dry of fresh mass?? Again an important detail left out

L163: seeds placed in plastic bags?? For how long? Why? Need justification here

L171: subsequent test? What subsequent test? Move this to where you talk about those tests.

L177: GARWOOD give the name! need to list and explain these seedling types from the onset.

L185: Phylocomm—using which base tree??

L205- 213; need a better/ clearer distinction between the parts of the study (greenhouse vs. other). What question does each part address?

L209: each response variable? Are these separate models. Models need to be described sufficiently to permit their reconstruction.

L216 of the species we observed in the field

L218: no mention of chi-squared tests in the methods? Why was this done? Which question does it address

L 224 Not clear what the growth form distinctions are here. They were not introduced prior.

L226-229: why are the phyloGLS results in the supplementary information and the rudimentary explanatory histogram plots in the results section. Put the most important figs/ info up front

Fig3; what is the point of fig 3. I looked at it a while and could not get the main point it is trying to convey.

L257: lower germination --- need stats to determine if actually lower.

L265: table 1 is too big and not organized well. Are these different models? What is the model structure? What are the models AIC?

L257: bootstrap? This belongs in methods.

L283: were excluded? Why? Need justification in methods.

L285-290: can you even trust the greenhouse data if they were all germinated in the same environmental conditions> need to better discuss methods and limitations.

L303: sorry but 0.1g is not a heavy seed. Even your heaviest seed is not a heavy seed in the grand scheme of things. Think about a mango seed -> hundreds of grams. Need to think about your data in the grand scheme of seed morphology

L362: light? Is light every limiting in TDFs?

L396: “but in these correlations, there was a strong phylogenetic signal” Did you want to take phylogeny into account and then test for things (i.e. account for phylogeny). If so, you need PIC – phylo-independent contrasts.

Lasky, Jesse R., María Uriarte, and Robert Muscarella. "Synchrony, compensatory dynamics, and the functional trait basis of phenological diversity in a tropical dry forest tree community: effects of rainfall seasonality." Environmental Research Letters 11.11 (2016): 115003.

Muscarella, Robert, et al. "Life‐history trade‐offs during the seed‐to‐seedling transition in a subtropical wet forest community." Journal of Ecology 101.1 (2013): 171-182.

Poorter, Lourens, and Lars Markesteijn. "Seedling traits determine drought tolerance of tropical tree species." Biotropica 40.3 (2008): 321

Reviewer #2: This paper addresses the variation of seed germination and other seedling traits in relation to functional traits and attempts to relate them to regeneration patterns in a tropical dry forest in Michoacan México. While the study provides new and detailed information on several tropical dry forest species, I believe it failed to elucidate the implications of the results and to contrast them with global tropical dry forests or other tropical ecosystems. For instance, what are the expected implications of the observed patterns for successional trajectories? How could the results de used for management and restoration efforts? I believe that the discussion section could be much improved. The conclusion section is yet again another abstract on the results. Another big concern is on the extremely low R2 values on the regression and whether or not strong conclusions can be drawn from these results. Particular concerns are listed below.

For all graphs in figure 3, we should have post hoc tests and significant differences should be marked in the graphs.

The discussion seems to be more like an explanation of the results and less of a discussion where the study is situated in a worldwide context and where the implications and relevance of the results is highlighted

I was unable to access any of the supplementary materials

Line 164- So, the soil varied from one species to another? I think this is a problem of confused effects from germination and soil quality. Was the soil for all plastic bags the same? Species occurring in a given place is not an indication of optimal conditions and therefore, I'm not at ease with the idea about the soil not being the same for all germination beds. If this is not the case, please clarify.

Line 280- The authors mention a strong phylogenetic signal between Seedling type and seed size, and while the P value shows significance, the R2 value is 0.08, meaning it only explained .8% of the variance

Line 286- Again, the R2 values are extremely low

Line 293- I worry that most of these “strong phylogenetic signals” are based on very low R2 values, even if P- values show significance.

Table 1- follow the same order in the table and figures; growth forms, dispersal syndromes and dispersal phenology

Figure 1- The axes must be fixed to adjust the precipitation units to be double the temperature units so that the months under hydric stress can be visualized.

Figure 3- I found this figure to be quite overwhelming. Since each part of the graph is referred to in a different section of the text, I see no need to combine it al in a very large and complicated graph.

Reviewer #3: The main objective of the work presented by Cortés and collaborators is show the relationship among the germination characteristics and the types of seedlings with some seed and seedling traits such as growth form, seed mass, dispersal syndrome, and dispersal phenology. The topic addressed is of theoretical interest for the field of plant ecology and, in addition, provides valuable information on different aspects of reproductive / regenerative biology of a significant number of species of tropical dry forest.

The manuscript is well written and presents a clear thread in its development. In general, the methodology and the analyzes used are adequate as well as the bibliography cited. Below I mention only some minor aspects that I think may be useful for the authors.

Study area. What was the approximate size of the selected area? The selected species are from mature forests or belong to another type of plant community? (e.g. secondary forest).

Seed Germination

Line 164-165. Say: “The substrate was obtained from the site where the species occurred.” The substrate was mixed with sand? Also, how were the trays watered? What was the incubation temperature?

Discussion

Line 301-304. Say: “In contrast to the assumption that anemochorous species produce light seeds [48], we found that these species had large and heavy seeds (in average 0.10 g), particularly pterochorous diasporas related to woody species.” Do the authors have an explanation for this unexpected result?

Line 332. Bet-hedging is more associated with desert climates, here a very seasonal climate is observed with a clear differentiation between seasons (well predictable). Probably it is only a dormancy mechanism to avoid the unfavorable period.

Conclusion

This section is a bit repetitive with the discussion. It could be deleted and keep the paragraph: “However, further study is necessary to fully understand the establishment process of species associated with different seedling types under natural conditions. This information could help understand the dynamics of natural regeneration, a highly important topic, particularly in tropical dry forests, which are considered the most threatened biome throughout the world.” as the closing of the article.

6. PLOS authors have the option to publish the peer review history of their article (what does this mean?). If published, this will include your full peer review and any attached files.

Reviewer #1: No

Reviewer #2: No

Reviewer #3: No

---

## [Author Response · Author response to Decision Letter 0]

17 Dec 2019

Reviewer #1

Guillermo Ibarra-Manríquez et al. “Functional correlations of seed germination and seedling types

Summary: Ibarra-Manríquez et al. seek to uncover the complex nature of species regeneration trade-offs of the plants in a tropical dry forest community in Michoacán Mexico. They employ a perspective that examines the phenology of trees observed in the field, some greenhouse germination experiments and species characteristics (dispersal syndrome, seedling type, etc.) to connect the dots and provide insight into the regeneration differences and of the variation plant strategies that are present in that forest. Unfortunately, the manuscript lacks a sufficiently coherent organization or main message for me to recommend it for publication. Specifically, the questions are not as clearly formulated as they need to be and it is not clear how all of the pieces fit together to get at the questions. Overall, I found the paper difficult to read and understand. I found myself questioning about various details that would seem important to the study and in my opinion, the literature on the phenological and regeneration ecology of tropical dry is not referenced or incorporated well enough. To that tune, I make several suggestions that aim to improve the manuscript:

Main comments:

Comment 1. Seedling type? The mention of seedling type does not appear until line 80, and then again on lines 90 and 92. In all three cases, it is introduced in passing. Seedling types are not explained until the methods starting on lines 173. If this is a key focus of the paper, where the authors aim to make a contribution, then both the formal definitions of each seedling type (with examples) and the logic behind their relationships to species phenology and regeneration dynamics need to be explained and emphasized from the beginning. When I read it at first (both on like 90 and 92) I was wondering what seedling type meant.

Response 1. Taking into account the reviewer's suggestion, in lines 75-79 of the new version, we describe the traits to classify the seedling types and one example of their role in the tropical dry forest. Besides, we describe seedling types in lines 172-176.

Comment 2. Phylogenetics – It is true, as the author state throughout, that phylogeny (i.e. plant lineage) is a predictor of functional trait values. For example, wood density and seed mass have phylogenetic structure. I do not think it is enough to simply point (e.g. L 69) out the fact that phylogeny matter, as that is common knowledge at this point. If you do discuss the effect of plant lineage, then you should state exactly how phylogeny structure the trait of interest. For example, higher order Angiosperms tend to have narrower root diameters than do lesser-evolved basal angiosperms and gymnosperms. You can move from that starting point to form some hypotheses about how things might be operating from there.

Response 2. We agree with the first comment of the reviewer and in lines 62-67 we empathize that many plant traits are not phylogenetically independent, and some ecological relationships can be determined by the evolutionary history of the species. Nevertheless, we cannot generate a hypothesis about the evolution of traits, since in this study we only evaluate the phylogenetic signal in the correlations between traits and not evolution traits per se.

Comment 3. A second point about phylogenetics – In most of your analyses you seem to be looking for phylogenetic structure in the variables of interest. But this is not well justified. I could see many of these questions going the other way, as in, do relationships exist after accounting for phylogenetic structure in the data. For example is seed mass related to seedling type (after controlling for differences in seed mass due to plant lineage), which I think is what you are getting at. In that case, you would use phylogenetically independent contrasts, not pGLMs. Either way, you need to justify the use of the statistics and link them clearly back to questions that intended to answer and why and how they do so.

Response 3. We agree with this commentary. In lines 85-92 of the current version we made this important addition. In relation with selection of one or another phylogenetic statistical method, this depends on the type of data (Garamszegi, 2014). In our case, some of the response variables are binary (e.g. seedling type, germination) and many of explanatory variables are discrete (e.g. growth form, dispersal syndrome). Instead, phylogenetically independent contrasts were designed for continuous data. Therefore, we implemented phylogenetic logistic regression and phylogenetic generalized least squares analysis since they are appropriate analysis for binary data and discrete explanatory variables, respectively.

Reference.

Garamszegi LZ. 2014. Modern phylogenetic comparative methods and their application in evolutionary biology. Springer, Berlin, Heidelberg.

Comment 4. Hypotheses: you have none. Did you have some expectations when you started? List them. If you did not then at least state what the literature should propose (ie the starting point as mentioned above). The questions also run together throughout and are not clearly differentiable (See point 1).

Response 4. We do not totally agree with the reviewer but we modified this last part of the introduction. Although we do not propose a hypothesis, we have now three objectives coupled with a set of predictions based on literature (lines 92-104) and are revealed previously in this section of the manuscript.

Comment 5. “Functional traits?” only one of the things you call functional traits in your introduction is in fact a functional trait, based on the common definition of a function trait (Violle et al 2007 Oikos 116:882-892): ‘‘Functional traits’’ are defined as morpho-physio-phenological traits which impact fitness indirectly via their effects on growth, reproduction and survival, the three components of individual performance.” The one that is a true functional is seed mass (for reasons you more or less explain in your paper). The others are not. Phenology is phenology. The same goes for dispersal syndrome. Along the same lines (as I side not), I found the use of the terminology “germination response” cumbersome and overly complex. Was it not simple germination (whether you could grow the seedling or not)? Why not just use germination rates, and where nothing germinated the rate is zero? The germination rates and other demographic estimates are just that, indicators of species life-histories. They are not traits; they are not functional for anything. They are estimates for how the species perform (i.e germinates/ establishes) in the environment.

Response 5. We thank the reviewer for this valuable comment. Consequently, in the new version, we refer to the traits studied as plant life-history traits. In relation with her/his last part of the comment, we change the term “germination response” by “germination” along of the manuscript.

Comment 6. Models: Models need to be sufficiently explained and described using sufficient statistical detail to permit their re-creation from the data (which should be freely available). I did not find that to be the case in this work. Models were vaguely described (i.e no model formulae were given), and I found the result tables difficult to interpret. In many cases, it was unclear if multiple models were fit with multiple response variables or not. If multiple models are fitted, then each needs to be discussed as a separate entity, even if the co-variate structure of the models it the same.

Response 6. Again, we agree with the proposal of the reviewer. In this new version, we specify that using a phylogenetic generalized least squares model in order to test the relationship between seed mass and other plant life-history traits. Moreover, we emphasize the implementation of individual models to evaluate the relationship between germination (response variables) and other attributes (explanatory variables) (lines 187-190). Finally, we add as part of the main text a table with the constructed models and complete results of analyzes (Table 1 in the current version).

Comment 7. Fitting the pieces together. / Organization. Have you ever heard anyone say “organization is half the battle?”…. well it’s true, in fact, it’s actually more than half the battle. It is imperative that information in scientific journals is organized in a fashion that permits its easy consumption (reading/referencing). There are conventions on how to lay out the figures and tables for a paper, similarly, there are conventions for the organization the flow of ideas and the results in the prose of the manuscript body. This work needs an overhaul/ massive re-organization of the information and structure of the writing. Subheadings were used, but do little to guide the reader through the cluttered mass of information. It is unclear which part of the study (the fieldwork vs. the greenhouse) the writers are referencing in many places. Also, concepts and ideas/ terminology need to be consistently used, defined from the beginning and organized in a manner that flows logically. Tables could be used in places where they are not (L116-127). And many of the main figures belong in the supplement and vice versa. The main findings of the pGLS models, which are the backbone of the study, appear in the supplement. I found these models impossible to reconstruct in my mind given the way they were presented.

Response 7. We acknowledge this significant comment. In the new version, we add two subheadings to make a distinction between the data record of plant life-history traits and germination. In relation to the Table suggested by the reviewer, we give a vegetation type description of the study area (lines 111-123). Also, in the subsection of seed germination and seedling types, we describe that seed germination experiments were performed in a shade house and indicate details about experiments (lines 157-164). Finally, we include complete results of PGLS models in Table 1, and we also made some changes in the presentation of Fig. 2 to showing germination patterns and seed mass on the phylogenetic tree.

Line comments:

Comment 8. L18: phases for natural regeneration. Is succession guided? If it is I have a hard time believing that it is guided by seed germination… I guess a case could be argued there but start with concrete clear sentences to draw the reader in, not tangential, controversial claims. Be cognizant of the words you choose.

Response 8. We agree with this comment by the reviewer, and we have rewritten this sentence (lines 17-19).

Comment 9. L20: there is an important intraspecific variation? ?? check grammar and meaning

Response 9. In line 18 we refer to the interspecific variation at the community level.

Comment 10. L 21: Delete, however; variation in seed germination

Comment 11. L22. Regeneration and succession 

Response. We have rewritten the text in lines 19-22. 

Comment 12. L 23: evaluated integrally?? Check Grammar

Response 12. We have deleted the word “integrally”.

Comment 13. L24: response? Response to what?? Perhaps germination percentage, lag time and rate?

Response 13. We have taken into account the suggestion by the reviewer and we have rewritten the sentence (line 24).

Comment 14. L25 I like dispersal method or mode better than syndrome.

Response 14. We do not agree with the suggestion of the reviewer. We determined a set of characteristics of diaspores following the van der Pijl (1972) proposal of dispersal syndromes, and we then complemented this information with field observations.

Comment 15. L29: “we detected a strong phylogenetic signal” GIVE THE NUMBERS! Stats! Previous sentences in abstract say nothing of phylogenetics. How and why are you doing this? Need a preview.

Response 15. This comment of the reviewer is correct. Nevertheless, we do not give any statistical value throughout the abstract because in this section of similar studies generally values are mentioned only if the relationships found shows a phylogenetic signal or not. Besides, abstract was rewritten from this line to the end (lines 31-38).

Comment 16. L30: species that dispersed seeds

Response 16. We change this sentence (line 27).

Comment 17. L31: germination was 30 percent compared to 50 percent for those that dispersed seeds at the end of the dry season

Response 17. The sentence was delete because we have rewritten the text (lines 27-30).

Comment 18. L 33: germination lag time.

Response 18. We made the addition suggested (line 31).

Comment 19. L34: “showed an important relationship” what was the relationship??!

Response 19. In lines 31-32, we added that germination lag time and germination rate were negatively related to the seed mass.

Comment 20. L36: Give number here. What is the trend? Big seeds from shade tolerant species have what type of seedling.

Response 20. In lines 32-34, we added that predominant seedling type in the community was the phanerocotylar epigeal with foliaceous cotyledons (56 %), mainly associated to light seeds and herbs species.

Comment 21. L41: are considered some of the most threatened ecosystem in the world. This seems like a weak ending to the abstract that is not at all related to the findings of the paper. What is your take-home message? Drive it home here, instead of this conservation pleading weak finish. Know your audience. Give them what they need to know.

Response 21. We agree with the reviewer and we have rewritten the text (lines 34-38).

Comment 22. L46: aka the seed to seedling transition (see the paper by Muscarella et al and references within)

Response 22. We made the change suggested by the reviewer, and we added the citation of Poorter (2007) (line 41).

Comment 23. L47: in seasonally-dry ecosystems? … seasonally-available water resources? (such as?)

Response 23. In line 41 we add “seasonally dry ecosystems”. 

Comment 24. L48; wet season?

Response 24. We change “growth season” to “wet season” (line 43).

Comment 25. L58: “is frequently found some innate” Grammar

Response 25. We have rewritten the text of line 54.

Comment 26. L59: postpones?? Germination

Response 26. We change the word using “to delay germination” (line 55).

Comment 27. L66: seed mass is the only functional trait here, the others are more life history characteristics or species demographic attributes.

Response 27. We agree with this suggestion of the reviewer, and throughout the text, we change "functional traits" by “plant life-history traits”.

Comment 28. L69: what is the phylogenetic trend of this trait?

Response 28. We thank the reviewer for the comment. We have rewritten the text (lines 62-67).

Comment 29. L 74: see Lasky et al. Environment Research Letters. Wood Specific Gravity and taproot presence were shown to be important traits in a Puerto Rican dry forest.

Response 29. Perhaps the reviewer misunderstood our sentence. In lines 68-69, we refer to germination studies in tropical dry forest communities, however, the work of Lasky et al. (2016), addresses the phenological responses and functional traits of trees.

Comment 30. L 75: This statement is not true! See Poorter & Markesteijn 2008/

Response 30. We have written the text of lines 71-72 and added the reference suggested by reviewer and Muscarella et al. (2013).

Comment 31. L80: what is seedling type? Need definition

Response 31. We now mention that seedlings types were defined from cotyledon morphology, considering their position, exposure, and function and we have also indicated the references where they are defined more extensively (lines 75-77). Short description of the diagnostic attributes of each seedling type found in our study is found in lines 172-176. 

Comment 32. L83-85: This sentence is too vague

Response 32. We have rewritten the text of sentences (lines 80-82).

Comment 33. L86-89: so what? What is the point here? Why does it matter? Do you focus on more than just trees? That is not clear? Needs better logical flow

Response 33. In these lines, we mention that in the tropical dry forest, the information on seed germination and seedling traits is scarce and comes mainly from studies developed in certain tree species; as an alternative, our study includes other growth forms, and therefore information generated can provide a more complete picture about community patterns in this ecosystem (lines 80-85).

Comment 34. L93: move this up in the introduction. Get to this point first, which forms a solid jumping off point for this study.

Response 34. We made the change indicated by the reviewer (lines 66-67).

Comment 35. L96 We predicted

Response 35. We made the change (lines 92-93).

Comment 36. L103: seed coat can vary with dispersal syndromes… how?

Response 36. Baskin and Baskin (2001) suggest that germination patterns may be related to dispersal syndrome; for example, some seeds dispersed by animals have an impermeable coat that requires mechanical or chemical scarification for germination, while seeds dispersed by wind generally lack physical dormancy and germinate rapidly (lines 98-100).

Comment 36. L94-107: These hypotheses are circular. Use the point about seed mass as a foundational starting point to flesh out questions about the seedling type and underlying demographics of them. It is not a good place to start your hypotheses, which I think are more about the other things, like how seedling type relates to seed mass and demographics, correct?

Response 36. We recognize this critical and valuable comment. In the revised version, we framed our study with three goals (lines 88-92): i) characterized the dispersal phenology, dispersal syndromes, growth forms, and seed mass at the community level; ii) determined to what extent the interspecific variation in seed germination and seedling types can be explained by plant life-history traits and phylogeny; iii) evaluated whether seedling types had different germination patterns. Then, we give a set of predictions derived from each question, based on the literature and justified in the theoretical framework of introduction (lines 92-104).

Comment 37. L117-127: list these in a table – give families and taxonomic authorities.

Response 37. We thank the reviewer for this comment. Nonetheless, we do not totally agree with him/her because in these lines we only give a vegetation type description and their frequent species (lines 113-123) but in order to face this comment, we provide a new Table (supplementary material, S1 Table), where species (which include taxonomic authorities) and their families and growth form are provided.

Comment 38. L121: Stenocereus – since when is cactus a tree??

Response 38. Columnar cacti were considered as an arboreal growth form species (definition of this growth form can change according to the individual author). Therefore, in line 113 we change “tree” to “arboreal species”.

Comment 39. Fig 1: Need to label y-axes. Where is the climate station in relation to the study site?

Response 39. We made the changes indicated by the reviewer in the Figure 1 (line 129).

Comment 40. L136: how were the observations made? Using binoculars? Seed traps? These are very important details. They need to be included

Response 40. We indicate in the sentence that phenological data records were based on direct observations of the species and the measurements of the transect used (lines 132-134).

Comment 41. L 139: why? What does this represent?

Response 41. We have rewritten the text of lines 135-137.

Comment 42. L146: Following the proposed classification by __________. Give name and year

Response 42. We have inserted the reference of van der Pijl (1972) (Line 144).

Comment 43. L154: size? Mass is not size? Did you measure the dry of fresh mass?? Again an important detail left out

Response 43. We mentioned “to obtain the fresh mass of seeds” (line 150).

Comment 44. L163: seeds placed in plastic bags?? For how long? Why? Need justification here

Response 44. We rewrote this paragraph and we now describe the method for this part of the study (lines 157-161). In lines 163-164 we mentioned that germination was recorded every third day for a maximum period of 60 days.

Comment 45. L171: subsequent test? What subsequent test? Move this to where you talk about those tests.

Response 45. We made the change suggested by the reviewer and we have moved this sentence to lines 198-199.

Comment 46. L177: GARWOOD give the name! need to list and explain these seedling types from the onset.

Response 46. The seedling types names are based on the cotyledon traits, which are explained in lines 172-176.

Comment 47. L185: Phylocomm—using which base tree??

Response 47. We specified that the super tree used was the R20120829 (line 179).

Comment 48. L205- 213; need a better/ clearer distinction between the parts of the study (greenhouse vs. other). What question does each part address?

Response 48. We thank the reviewer for this comment. We reorganized the data analysis section and ordered the study questions in the introduction to clarity each section of the manuscript.

Comment 49. L209: each response variable? Are these separate models. Models need to be described sufficiently to permit their reconstruction.

Response 49. We rewrote the paragraph (lines 203-213), and in Table 1 of the current version, we placed the complete models to permit their reconstruction.

Comment 50. L216 of the species we observed in the field

Response 50. We did not make the addition suggested by the reviewer since it is clear in the text that all the phenological observations were carried out in the field.

Comment 51. L218: no mention of chi-squared tests in the methods? Why was this done? Which question does it address. 

Response 51. As suggested by the reviewer, we included in the methods section the chi-squared test (lines 186-187).

Comment 52. L 224 Not clear what the growth form distinctions are here. They were not introduced prior. 

Response 52. In lines 199-200, we have added the growth forms considered in the analysis.

Comment 53. L226-229: why are the phyloGLS results in the supplementary information and the rudimentary explanatory histogram plots in the results section. Put the most important figs/ info up front

Response 53. According to the reviewer’s suggestion, in the current version, we added in the main text the results of the PGLS models (Table 1).

Comment 54. Fig3; what is the point of fig 3. I looked at it a while and could not get the main point it is trying to convey.

Response 54. We agree that Fig. 3 is a very large and complicated graph. So, we decided to delete this figure and, instead, add Table 1 with the complete results of PGLS models.

Comment 55. L257: lower germination --- need stats to determine if actually lower.

Response 55. In the current version, we added Table 1 with the values of the correlations between the attributes.

Comment 56. L265: table 1 is too big and not organized well. Are these different models? What is the model structure? What are the models AIC?

Response 56. We added the models tested in Table 2, and also reorganized their content.

Comment 57. L257: bootstrap? This belongs in methods.

Response 57. We made the addition suggested by the reviewer (lines 208-209).

Comment 58. L283: were excluded? Why? Need justification in methods.

Response 58. In lines 203-204 we mentioned “Given the low frequency of cryptocotylar seedlings with reserve cotyledons, they were excluded from the analysis”.

Comment 59. L285-290: can you even trust the greenhouse data if they were all germinated in the same environmental conditions> need to better discuss methods and limitations.

Response 59. We agree with this valuable comment, and in the current version, we discuss the methods and limitations of our study in different paragraphs of the manuscript (e. g. lines 340-345, 385-388).

Comment 60. L303: sorry but 0.1g is not a heavy seed. Even your heaviest seed is not a heavy seed in the grand scheme of things. Think about a mango seed -> hundreds of grams. Need to think about your data in the grand scheme of seed morphology

Response 60. This sentence is not included in the present manuscript because we have rewritten this part of the discussion. 

Comment 61. L362: light? Is light every limiting in TDFs?

Response 61. In the tropical dry forest, light can be considered a limiting factor for seed germination and seedling development during the rainy season because canopy species (trees and lianas) produce leaves and therefore limit the amount of light entry to the understory (lines 349-352).

Comment 62. L396: “but in these correlations, there was a strong phylogenetic signal” Did you want to take phylogeny into account and then test for things (i.e. account for phylogeny). If so, you need PIC – phylo-independent contrasts.

Response 62. We have taken into account the reviewer's observation and rewrote the text of lines 390-392. However, we do not agree that Phylogenetically independent contrasts (PICs) are a unique method for taking into account phylogeny. Currently, PICs are only one of the methods that can be implemented in ecological analyzes to considerer species phylogeny (e.g. Garamszegi, 2014). The PICs are a not very flexible method since they only consider a model of traits evolution (Brownian motion model) and only continuous variables can be evaluated. Instead, PGLS models are more flexible in both aspects because they consider other evolution models e.g. Ornstein–Uhlenbeck and can include discrete and continuous predictors. Finally, phylogenetic logistic regression is a statistical method adequate for analyzing binary dependent variables (Ives and Garland, 2010). The complete references mentioned previously are:

Garamszegi LZ. 2014. Modern phylogenetic comparative methods and their application in evolutionary biology. Springer, Berlin, Heidelberg.

Ives AR, Garland T. 2010. Phylogenetic logistic regression for binary dependent variables. Syst. Biol. 59: 9–26.

Reviewer #2: 

This paper addresses the variation of seed germination and other seedling traits in relation to functional traits and attempts to relate them to regeneration patterns in a tropical dry forest in Michoacan México. While the study provides new and detailed information on several tropical dry forest species, I believe it failed to elucidate the implications of the results and to contrast them with global tropical dry forests or other tropical ecosystems. 

Comment 1. For instance, what are the expected implications of the observed patterns for successional trajectories? How could the results de used for management and restoration efforts? I believe that the discussion section could be much improved.

Response 1. We thank the reviewer for the comment. In the current version, we added the implications of our results in the dynamics of natural regeneration and for ecological restoration in dry tropical forests along the manuscript (e.g. lines 36-38, 393-395).

Comment 2. The conclusion section is yet again another abstract on the results. 

Another big concern is on the extremely low R2 values on the regression and whether or not strong conclusions can be drawn from these results. Particular concerns are listed below.

Response 2. We agree with this interesting comment, and in the current version, we have reworded the conclusion. In relation to the second part of the comment, in many phylogenetic comparative analyses it is recognized that the proportion of variance explained (R2) by statistical models is very low, frequently in the range 0.05–0.10. This low explanatory power is in spite of overall model significance or terms within models being statistically significant. The explanation is simply that in large datasets it is possible to detect very small effects. As comparative datasets become larger, it is possible to pick up statistically significant effects of ever weaker signals, yielding models with statistically significant terms yet low overall explanatory power (Freckleton, 2009).

Freckleton, RP. 2009. The seven deadly sins of comparative analysis. Journal of Evolutionary Biology, 22, 1367– 1375.

Comment 3. For all graphs in figure 3, we should have post hoc tests and significant differences should be marked in the graphs. 

Response 3. We delete Fig 3 and provide Table 1 with the values of the correlation coefficients between traits.

Comment 4. The discussion seems to be more like an explanation of the results and less of a discussion where the study is situated in a worldwide context and where the implications and relevance of the results is highlighted

I was unable to access any of the supplementary materials

Response 4. In the new version of the manuscript, we added in different sections of discussion the implications for regeneration trajectories and ecological restoration in the tropical dry forest. Supplementary materials were modified and they are easy to check because are Word files.

Comment 5. Line 164- So, the soil varied from one species to another? I think this is a problem of confused effects from germination and soil quality. Was the soil for all plastic bags the same? Species occurring in a given place is not an indication of optimal conditions and therefore, I'm not at ease with the idea about the soil not being the same for all germination beds. If this is not the case, please clarify.

Response 5. We thank the reviewer for the comment. In lines 158-160 of the current version, we indicate that the substrate used was the same for all species and was composed of sand and clay in a proportion 60:40, respectively.

Comment 6. Line 280- The authors mention a strong phylogenetic signal between Seedling type and seed size, and while the P value shows significance, the R2 value is 0.08, meaning it only explained .8% of the variance.

Comment 7. Line 286- Again, the R2 values are extremely low

Comment 8. Line 293- I worry that most of these “strong phylogenetic signals” are based on very low R2 values, even if P- values show significance.

Response 6. The PGLS analysis investigates the impact of one or several predictor variables on a single response variable while controlling for potential phylogenetic signal in the response (and, hence, non-independence of the residuals), and express the phylogenetic signal in lambda value, but this metric does not have a relationship to model fit (Mundry, 2014).

Mundry R. (2014) Statistical Issues and Assumptions of Phylogenetic Generalized Least Squares. In: Garamszegi L. (eds) Modern Phylogenetic Comparative Methods and Their Application in Evolutionary Biology. Springer, Berlin, Heidelberg

Comment 9. Table 1- follow the same order in the table and figures; growth forms, dispersal syndromes and dispersal phenology

Response 9. We have made the suggested changes by the reviewer (Table 2 in the current version).

Comment 10. Figure 1- The axes must be fixed to adjust the precipitation units to be double the temperature units so that the months under hydric stress can be visualized.

Response 10. In the current version of Fig 1, we added the axis legends, but we do not adjust the precipitation units since the graph loses clarity in the distribution of photoperiod and temperature data.

Comment 11. Figure 3- I found this figure to be quite overwhelming. Since each part of the graph is referred to in a different section of the text, I see no need to combine it al in a very large and complicated graph.

Response 11. We delete this figure, and instead, we added Table 1 with the complete results of PGLS models.

Reviewer #3: 

The main objective of the work presented by Cortés and collaborators is show the relationship among the germination characteristics and the types of seedlings with some seed and seedling traits such as growth form, seed mass, dispersal syndrome, and dispersal phenology. The topic addressed is of theoretical interest for the field of plant ecology and, in addition, provides valuable information on different aspects of reproductive / regenerative biology of a significant number of species of tropical dry forest.

The manuscript is well written and presents a clear thread in its development. In general, the methodology and the analyzes used are adequate as well as the bibliography cited. Below I mention only some minor aspects that I think may be useful for the authors.

Comment 1. Study area. What was the approximate size of the selected area? The selected species are from mature forests or belong to another type of plant community? (e.g. secondary forest).

Response 1. We thank the reviewer for this comment. In lines 112-113 of the current version, we mention that vegetation type is mostly secondary tropical dry forest with more than 50 years of abandonment after being used for agriculture. Besides, we now indicate the size of the study area (lines 133-134).

Seed Germination

Comment 2. Line 164-165. Say: “The substrate was obtained from the site where the species occurred.” The substrate was mixed with sand? Also, how were the trays watered? What was the incubation temperature?

Response 2. In lines 158-160, we mentioned that substrate was obtained from the study site and was composed of sand and clay in a proportion 60:40, respectively, and we used the same substrate for all species. Also, in lines 162-163, we mentioned that the substrate was watered every third day with a manual sprinkler. Finally, in this current version we mention that seed germination experiments were performed in the study site, inside a shade house with an 80% shade cover (lines 157-158).

Discussion

Comment 3. Line 301-304. Say: “In contrast to the assumption that anemochorous species produce light seeds [48], we found that these species had large and heavy seeds (in average 0.10 g), particularly pterochorous diasporas related to woody species.” Do the authors have an explanation for this unexpected result?

Response 3. Not really because the germination success in the communities is a complex phenomenon. In this new version, this paragraph was lost because this section was modified. 

Comment 4. Line 332. Bet-hedging is more associated with desert climates, here a very seasonal climate is observed with a clear differentiation between seasons (well predictable). Probably it is only a dormancy mechanism to avoid the unfavorable period.

Response 4. We agree with the reviewer but this paragraph also was lost because this section was rewrotte.

Conclusion

Comment 5. This section is a bit repetitive with the discussion. It could be deleted and keep the paragraph: “However, further study is necessary to fully understand the establishment process of species associated with different seedling types under natural conditions. This information could help understand the dynamics of natural regeneration, a highly important topic, particularly in tropical dry forests, which are considered the most threatened biome throughout the world.” as the closing of the article.

Response 5. We thank the reviewer for the comment, and in the new version, we have rewritten our conclusions.

---

## [Decision Letter · Decision Letter 1]

14 Feb 2020

PONE-D-19-15449R1

Disentangling the influence of ecological and historical factors on seed germination and seedling types in a Neotropical dry forest

PLOS ONE

Dear Dr. Ibarra-Manríquez,

Thank you for submitting your manuscript to PLOS ONE. After careful consideration, we feel that it has merit but does not fully meet PLOS ONE’s publication criteria as it currently stands. Therefore, we invite you to submit a revised version of the manuscript that addresses the points raised during the review process.

We would appreciate receiving your revised manuscript by Mar 30 2020 11:59PM. To enhance the reproducibility of your results, we recommend that if applicable you deposit your laboratory protocols in protocols.io, where a protocol can be assigned its own identifier (DOI) such that it can be cited independently in the future. For instructions see: http://journals.plos.org/plosone/s/submission-guidelines#loc-laboratory-protocols

We look forward to receiving your revised manuscript.

Kind regards,

Ines Ibáñez, Ph.D.

Academic Editor

PLOS ONE

Additional Editor Comments (if provided):

Given the comments from one of the reviewers and my own reading of the revised version my recommendation is that the authors try again to improve clarity of the methods section. I myself have no clear what is actually being analyzed, predictors and responses, and this is important because it determines the analytical methods. Right now data have been heavily transformed to fulfill assumptions but it is not clear if a simpler generalized models would have work equally well.

Reviewers' comments:

Reviewer's Responses to Questions

**Comments to the Author**

1. If the authors have adequately addressed your comments raised in a previous round of review and you feel that this manuscript is now acceptable for publication, you may indicate that here to bypass the “Comments to the Author” section, enter your conflict of interest statement in the “Confidential to Editor” section, and submit your "Accept" recommendation.

Reviewer #1: All comments have been addressed

Reviewer #4: (No Response)

2. Is the manuscript technically sound, and do the data support the conclusions?

Reviewer #1: Yes

Reviewer #4: Partly

3. Has the statistical analysis been performed appropriately and rigorously? 

Reviewer #1: Yes

Reviewer #4: No

4. Have the authors made all data underlying the findings in their manuscript fully available?

Reviewer #1: Yes

Reviewer #4: Yes

5. Is the manuscript presented in an intelligible fashion and written in standard English?

Reviewer #1: Yes

Reviewer #4: Yes

6. Review Comments to the Author

Reviewer #1: L 33: which mostly associate

This paper looks to be much improved. The figures have been improved. The organization is much better, and the language and focus have been polished to where it much more understandable. Most all of the comments were incorporated in the manuscipt to a satisfactory degree. I commend the authors on thorough revision

Reviewer #4: In this study, the authors examine the correlates of germination success in a tropical dry forest. Specifically, they examine how seed germinate rates and lag time can be explained by species’ seed dispersal phenology, seed traits, dispersal mode and evolutionary history. Early life history dynamics such as seed-to-seedling transition plays a key part in plant community ecology and this study adds to that body of knowledge. Moreover, as best as I know, most seed/seedling studies in forests have been done in wetter regions, so this study from dry forests can increase the representation of biomes in the literature on early life-stage dynamics of plants. The inclusion of different life-forms is also valuable.

However, I find that some of the arguments for expected relationships are not intuitively clear in terms of the biological significance. For example, in lines 93-94, are you saying that despite differences in phenology, species will germinate at the same time or that most plants would disperse seeds at end of dry season? Similarly, the link between dispersal syndrome, seed mass and lag time (line 99-100) needs more detail. It is also not entirely clear to me why seedling types matter. Please explain the ecological significance of understanding differences in seedling types and their correlation with life-history traits.

The introduction ends with what was done rather than why it was done. It would help the reader to have your objectives written out as questions or more investigative statements. I suggest distilling the arguments from the previous paragraphs into statements of the specific research questions or gaps that you addressed here. For instance, can you provide a one-line justification for why you characterized dispersal phenology, dispersal syndromes etc. In general, the introduction can be reorganized to present specific packets of information in the order that they lead up to the main questions so as to avoid being repetitive and have better flow.

The methods for shadehouse experiments need to be explained more clearly. Ten seeds per bag seem like a lot for the size of the bag. Did you collect all local soil from the same location? Did you apply any pesticide treatment to the soil to avoid enemy infestation? If you do not control for insects or fungal infection, please discuss whether enemies might have differentially affected the germination of species with different traits (e.g., seed size, seed coat thickness).

As pointed out by a previous reviewer, the models need to be better explained/described. Until the results, the reader does not know how many models have been run, how they were set up (response and predictors) and how was model performance/fit assessed and models compared. Clearly, you had a multiple models per response, but how were they different? Did you conduct post-hoc pairwise tests for the different categorical variables (corrected for multiple comparisons)?

I am not sure whether you tested for interactions among variables. I would like to see whether the patterns for seed mass and germination rates were driven by life-form or were the slopes similar within each life-form. Does the forward slash in Table 1 indicate an interaction? Why does the second model not have the main effects of the variables?

It would be helpful to see plots and predicted lines for some of the relationships tested. The paper seems a bit heavy on tables and lean on figures.

Line 42: it is not clear why seedling establishment is based on resource acquisition for all species. What about seed resources?

Line 67: you mean rates of emergence?

Line 85: can you briefly justify why studies from trees may not be generalizable to herbs?

Line 95: what is the difference between rate and percentage?

Line 99: needs citation: “thickness of the seed coat can vary with dispersal syndrome”

Line 136: briefly explain how the statistical procedure works

Line 138: ‘mature fruit’?

Line 162: what was the rationale for burying seeds this way?

Line 169: I am a bit confused about this. Did you dig up the seeds or use a fresh set that was scarified?

Line 170: why then did you scarify seeds? What was that used for?

Line 197: what assumptions were those?

Lines 203-213: why were predictors used separately?

7. PLOS authors have the option to publish the peer review history of their article (what does this mean?). If published, this will include your full peer review and any attached files.

Reviewer #1: Yes: J. Aaron Hogan

Reviewer #4: No

---

## [Author Response · Author response to Decision Letter 1]

20 Mar 2020

Reviewer #1

This paper looks to be much improved. The figures have been improved. The organization is much better, and the language and focus have been polished to where it much more understandable. Most all of the comments were incorporated in the manuscript to a satisfactory degree. I commend the authors on thorough revision.

Comment 1: L 33: which mostly associate

Response 1: We made the change suggested (lines 33-34).

Reviewer #4

In this study, the authors examine the correlates of germination success in a tropical dry forest. Specifically, they examine how seed germinate rates and lag time can be explained by species’ seed dispersal phenology, seed traits, dispersal mode and evolutionary history. Early life history dynamics such as seed-to-seedling transition plays a key part in plant community ecology and this study adds to that body of knowledge. Moreover, as best as I know, most seed/seedling studies in forests have been done in wetter regions, so this study from dry forests can increase the representation of biomes in the literature on early life-stage dynamics of plants. The inclusion of different life-forms is also valuable. However, I find that some of the arguments for expected relationships are not intuitively clear in terms of the biological significance.

Comment 1. L: 93-94, are you saying that despite differences in phenology, species will germinate at the same time or that most plants would disperse seeds at end of dry season? 

Response 1. We rewrite this sentence and explained that seed germination would be more frequent in species that disperse their seeds at the end of the dry or beginning of the rainy season (lines 94-96).

Comment 2. Similarly, the link between dispersal syndrome, seed mass and lag time (line 99-100) needs more detail. It is also not entirely clear to me why seedling types matter. Please explain the ecological significance of understanding differences in seedling types and their correlation with life-history traits.

Response 2. We agree with these valuable comments and in the current version we specified that since lag time and rate of germination can vary with the thickness of the seed coat, we expected that germination to be faster in seeds with a thin coat of wind-dispersed species, than in seeds with a thick coat of animals-dispersed species (lines 100-101). Besides, we added that the functional morphology of cotyledons (seedling types) is important on seedling establishment because seedling types are associated with other plant life-history traits. Particularly, we predicted that seedlings with foliaceous cotyledons would be more frequent in herbs (they have predominantly small seeds, therefore the lag time and germination rate would be faster), whereas seedlings with reserve cotyledons would be more frequent in woody species (associated more often with large seeds and, therefore, the lag time and germination rate would be slower). (lines 101-108).

Comment 3. The introduction ends with what was done rather than why it was done. It would help the reader to have your objectives written out as questions or more investigative statements. I suggest distilling the arguments from the previous paragraphs into statements of the specific research questions or gaps that you addressed here. For instance, can you provide a one-line justification for why you characterized dispersal phenology, dispersal syndromes etc. In general, the introduction can be reorganized to present specific packets of information in the order that they lead up to the main questions so as to avoid being repetitive and have better flow.

Response 3. The reviewer’ comments are correct. In this modified version we write our objectives as questions (lines 90-93).

Comment 4. The methods for shadehouse experiments need to be explained more clearly. Ten seeds per bag seem like a lot for the size of the bag. Did you collect all local soil from the same location? 

Response 4. We did not have a problem with the size of the bag, because we use black polythene bags of 20 × 15 × 30 cm, i.e., we sowed ten seeds in an area of 300 cm2. In relation with the second question, we added in the manuscript that soil was collected exclusively from the same place (lines 167-168).

Comment 5. Did you apply any pesticide treatment to the soil to avoid enemy infestation? If you do not control for insects or fungal infection, please discuss whether enemies might have differentially affected the germination of species with different traits (e.g., seed size, seed coat thickness).

Response 5. We specified now that in order to germinate the seeds in the most similar conditions to occurs naturally, which was one of the objectives to developt in this research, we do not apply any pesticides or fertilizers to the soil (lines 169-170). We had not any problems in the seed germination experiments related with pests or predators and thus, we do not mention nothing of this topic in the manuscript.

Comment 6. As pointed out by a previous reviewer, the models need to be better explained/described. Until the results, the reader does not know how many models have been run, how they were set up (response and predictors) and how was model performance/fit assessed and models compared. Clearly, you had a multiple models per response, but how were they different?

Response 6. We thank the reviewer for this comment. In lines 203-204 of the data analysis section, we mentioned that model selection was based on the Akaike information criterion (AIC). In the S2 Table, we describe all the models tested and show that the model selected was the one with the lowest delta AIC (i.e., the model with the best fit). In the heading of the Table 1, we show the final model and the results and indicate that lag time and rate of germination are the response variables (line 269).

Comment 7. Did you conduct post-hoc pairwise tests for the different categorical variables (corrected for multiple comparisons)?

Response 7. We do not conduct a post-hoc pairwise test for the categorical variables. For each level of categorical variables, we reported the estimated coefficients (β) and the p-value of the pgls models tested.

Comment 8. I am not sure whether you tested for interactions among variables. I would like to see whether the patterns for seed mass and germination rates were driven by life-form or were the slopes similar within each life-form. Does the forward slash in Table 1 indicate an interaction? Why does the second model not have the main effects of the variables? It would be helpful to see plots and predicted lines for some of the relationships tested. The paper seems a bit heavy on tables and lean on figures.

Response 8. All these comments are interesting. First, we do test the interactions among the predictors. For example, the best fit model to explain germination lag time included dispersal syndrome and growth form in interaction with seed mass as predictors. In the second model, the germination rate was explained by dispersal syndrome and growth form, both in interaction with seed mass. Second, in Table 1, we added that forward slash indicates the interactions between traits (we indicate this relation in the lines 270-271). Third, we believe that Table 1 shows clearly and synthetically the effects of explicative variables on response variables. A acknowledge that a figure can be visually useful, but the statistics values cannot be displayed completely. To resolve this controversy, we made the figure suggested by the reviewer and it will appear as supplementary material in the current version of the manuscript.

Comment 9. Line 42: it is not clear why seedling establishment is based on resource acquisition for all species. What about seed resources?

Response 9. In order to be clearer, in the current version we specified that successful seedling establishment depends on the seed resources and rapid acquisition of available environmental resources (line-42).

Comment 10. Line 67: you mean rates of emergence?

Response 10. Thanks for this observation; we inserted the word emergence (line 67).

Comment 11. Line 85: can you briefly justify why studies from trees may not be generalizable to herbs?

Response 11. We did it. In lines 83-84 we specified that trees have life-history strategies different from other growth forms supported by three references.

Comment 12. Line 95: what is the difference between rate and percentage?

Response 12. Because germination rate is calculated from the slope at the inflection point of the curve of germination, this metric includes percentage plus velocity of germination.

Comment 13. Line 99: needs citation: “thickness of the seed coat can vary with dispersal syndrome”

Response 13. We made the change suggested (inserted reference 11).

Comment 14. Line 136: briefly explain how the statistical procedure works

Response 14. We thank the reviewer for this suggestion. In lines 141-143, we explained that to calculate the mean angle (ā), months were converted into angles (e.g., January corresponds to 15°, February to 45° and so, and that the statistical significance of ā was determined using Rayleigh’s test (z).

Comment 15. Line 138: ‘mature fruit’?

Response 15. This observation is correct; we made the change suggested (line 139).

Comment 16. Line 162: what was the rationale for burying seeds this way?

Response 16. In lines 172-174 we mentioned that each seed was buried at a depth of approximately twice the seed width to prevent dessication of seed and besides, to facilitate its emergence.

Comment 17. Line 169: I am a bit confused about this. Did you dig up the seeds or use a fresh set that was scarified?

Response 17. We thank the reviewer for this comment. In lines 179-180, we specified that pregerminative treatments were applied to a new set of the seed of those species in which did not occur germination.

Comment 18. Line 170: why then did you scarify seeds? What was that used for?

Response 18. In the methods, we separated into two sections the seed germination experiment and the seedling types characterization. This experiment allowed us to identify which species not germinate and so, require some treatment to induce the germination and obtain the seedling (lines 179-181).

Comment 19. Line 197: what assumptions were those?

Response 19. We acknowdledge the reviewer's comment. We rewrote the text in lines 207-208: the germination percentage data were arcsine transformed to be included in the analyzed models.

Comment 20. Lines 203-213: why were predictors used separately?

Response 20. We related predictors separately because they are categorical variables, and the phylogenetic logistic regression analysis cannot form interactions between these predictors. Therefore, in Table 2, we report the coefficients for all levels of categorical predictors.

---

## [Editor Report · Decision Letter 2]

26 Mar 2020

Disentangling the influence of ecological and historical factors on seed germination and seedling types in a Neotropical dry forest

PONE-D-19-15449R2

Dear Dr. Ibarra-Manríquez,

We are pleased to inform you that your manuscript has been judged scientifically suitable for publication and will be formally accepted for publication once it complies with all outstanding technical requirements.

With kind regards,

Ines Ibáñez, Ph.D.

Academic Editor

PLOS ONE

Additional Editor Comments (optional):

The authors have successfully addressed the second round of comments made by the reviewers.
---

## [Editor Report · Acceptance letter]

2 Apr 2020

PONE-D-19-15449R2 

Disentangling the influence of ecological and historical factors on seed germination and seedling types in a Neotropical dry forest 

Dear Dr. Ibarra-Manríquez:

I am pleased to inform you that your manuscript has been deemed suitable for publication in PLOS ONE. Congratulations! Your manuscript is now with our production department. 

With kind regards,

on behalf of

Dr. Ines Ibáñez 

Academic Editor

PLOS ONE